# Residual Multiplicative Filter Networks for Multiscale Reconstruction

**Shayan Shekarforoush**[1,2]
shayan@cs.toronto.edu

**David B. Lindell**[1,2]
lindell@cs.toronto.edu

**David J. Fleet**[1,2,3]
fleet@cs.toronto.edu

**Marcus A. Brubaker**[1,2,4]
mab@eecs.yorku.ca

[1]University of Toronto   [2]Vector Institute   [3]Google Research   [4]York University

## Abstract

Coordinate networks like Multiplicative Filter Networks (MFNs) and BACON offer some control over the frequency spectrum used to represent continuous signals such as images or 3D volumes. Yet, they are not readily applicable to problems for which coarse-to-fine estimation is required, including various inverse problems in which coarse-to-fine optimization plays a key role in avoiding poor local minima. We introduce a new coordinate network architecture and training scheme that enables coarse-to-fine optimization with fine-grained control over the frequency support of learned reconstructions. This is achieved with two key innovations. First, we incorporate skip connections so that structure at one scale is preserved when fitting finer-scale structure. Second, we propose a novel initialization scheme to provide control over the model frequency spectrum at each stage of optimization. We demonstrate how these modifications enable multiscale optimization for coarse-to-fine fitting to natural images. We then evaluate our model on synthetically generated datasets for the the problem of single-particle cryo-EM reconstruction. We learn high resolution multiscale structures, on par with the state-of-the art. Project webpage: https://shekshaa.github.io/ResidualMFN/.

## 1 Introduction

Coordinate networks have emerged as a powerful way to represent and reconstruct images, videos, and 3D scenes [1–3], and for solving challenging inverse problems such as 3D molecular reconstruction for cryo-electron microscopy (cryo-EM) [4–6]. They typically take as input a point defined on a continuous, low-dimensional domain (e.g., a 2D position for images), and they output the signal value at that point (e.g., the color). In recently proposed architectures [7, 8], the frequency content of the represented signal can also be explicitly controlled. Such networks are motivated in part by the effectiveness of multiscale methods in image processing and 3D reconstruction.

Nevertheless, while current scale-aware coordinate network architectures offer some control over scale [8–10], they are not readily compatible with classical multiscale methods used for coarse-to-fine optimization, like pyramids [11, 12] or multigrid solvers [13]. State-of-the-art cryo-EM models, for instance, use *frequency-marching* to progressively estimate 3D density, beginning with a coarse structure, then adding finer structure at each iteration [14, 15]. Some networks use heuristics to empirically constrain the scale of network output for coarse-to-fine optimization, but they have no explicit constraints on the represented frequencies [9, 16, 17]. Others represent signals at multiple scales at inference time, but do not enable control of the frequency spectrum during training [8, 10].

36th Conference on Neural Information Processing Systems (NeurIPS 2022).

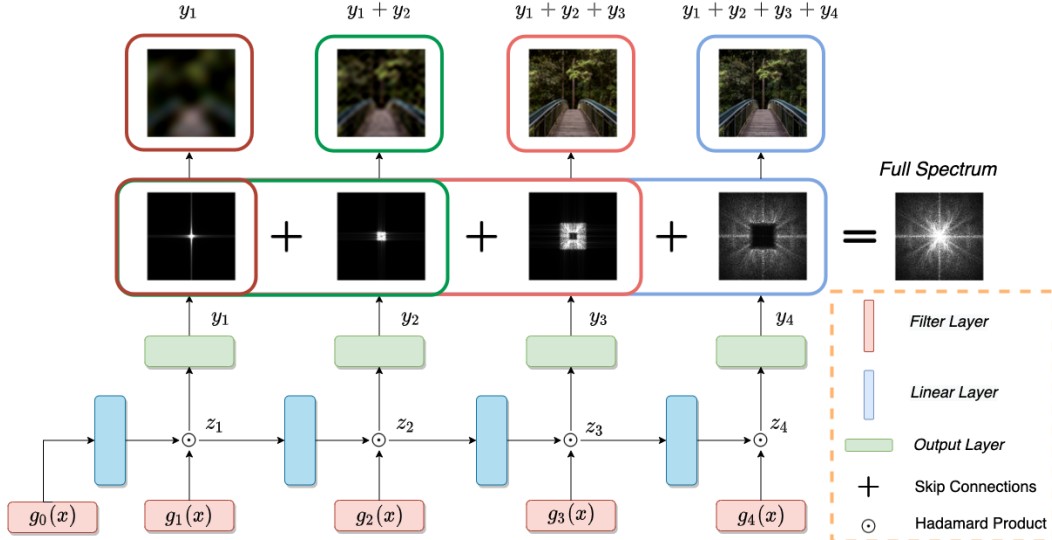

Figure 1: Overview of residual multiplicative filter networks. Skip connections efficiently combine low-frequency information from earlier layers with high-frequency information in later layers. With the proposed initialization scheme, each layer learns separate, increasing frequency bands.

Here, we introduce a new coordinate network architecture and training scheme that enables one to effectively control the frequency support of the learned representations during multiscale optimization, building on recent neural networks with an analytical and controllable Fourier spectra [7, 8]. For coarse-to-fine reconstruction, one can divide the training process into stages, each of which learns a single scale. Training starts at the coarsest resolution and then progressively adds finer scales. Naively employing this procedure with existing architectures will damage the signal reconstruction at coarser scales when fitting finer ones (e.g., see Fig. 3). To mitigate this, we modify MFNs by adding skip connections [18], so reconstruction at one scale naturally incorporates the learned structure at coarser scales. This alleviates the need to adapt previous layers in favor of fitting the finer grained reconstructions, effectively obviating the need to re-learn signal structure captured by coarser scales. Second, we derive an initialization scheme which explicitly introduces hierarchical gaps in the frequency spectrum of a new layer so that, by construction, its frequency support has a controllable degree of overlap. This requires the network to use the skip connections in order to fill the holes in the spectrum and ensures lower levels remain faithful to coarse scale representations of the signal.

We apply our technique to the inverse problem of single-particle cryo-EM reconstruction [6, 4, 14, 19] to determine the 3D structure of macromolecular complexes, like proteins, from collections of noisy 2D tomographic projections. This is a challenging non-linear inverse problem, closely related to multi-view reconstruction, requiring estimation of the pose and 3D structure of bio-molecules. Successful 3D reconstruction relies on coarse-to-fine methods, sometimes called frequency marching [20], to reduce the risk of becoming trapped in poor local minima. Our results demonstrate effective and efficient *ab initio* reconstruction of 3D structures with cryo-EM to high resolution.

This paper proposes a new architecture and training approach to enable powerful multiscale estimation techniques with coordinate-based networks. Specifically, we make the following contributions:

- We develop residual multiplicative filter networks, a new coordinate network architecture with a tailored initialization scheme for multiscale signal reconstruction and representation.
- We design fast and efficient coarse-to-fine training strategies for residual multiplicative filter networks that leverage the multiscale representation.
- We apply the proposed architecture and training strategy to cryo-EM reconstruction on two synthetic datasets and achieve final reconstructions competitive with the cryoSPARC [14], a state-of-the-art method.

## 2  Related Work

**Coordinate-based Networks.** Coordinate-based networks (e.g., [2, 3]) offer a memory-efficient, continuous function parameterization that can be flexibly trained to reconstruct 3D appearance [1, 21–25] and structure [26–32], including applications in biomedical imaging [33, 34] and cryo-EM [5, 4, 6]. Although originally formulated as fully-connected MLP architectures, new models have been proposed to improve performance and interpretability. For instance, one can use multiple small fully-connected networks to improve representational capacity [35–37], or combine coordinate-based networks with explicit feature grids to improve efficiency [38, 39], or facilitate generalization across shapes or scenes [40–43, 29, 44, 45]. Multiplicative Filter Networks (MFNs) replace the conventional MLP architecture with successive layers of Hadamard products and sine non-linearities [7]. Closest to our own work, MFNs have an analytical Fourier spectrum whose bandwidth can be explicitly constrained [8], improving the controllability and interpretability of the representation. Inspired by this work, we establish *Residual MFNs* (rMFNs) with specialized control over the Fourier spectrum to enable coarse-to-fine optimization.

**Multiscale Reconstruction.** Multiscale representations and reconstruction methods are fundamental concepts in signal and image processing. For example, wavelets are a fast and efficient multiscale signal representation [46, 47] useful for denoising, compression, communications, and optical flow estimation [48]. Multigrid frameworks are used for many differential equation solvers for physics simulations [13]. Gaussian and Laplacian pyramids [49, 11] have broad application to image processing and have, in part, inspired modern deep learning architectures [50].

Recent coordinate-based network architectures build on these fundamental concepts of multiscale signal representation and reconstruction. Some architectures use explicit feature grids at multiple scales to enable efficient training and inference for representing shape [39, 51] or rendering scenes [52, 38]. Multiscale fully-connected coordinate network architectures have been realized by progressively increasing the frequencies of positional encodings during optimization, for example, for bundle adjustment and neural scene representation [9, 17]. It is also possible to optimize separate networks with inductive biases towards low or high frequencies to improve shape representation [16]. Still, while these methods introduce multiscale training techniques, the architectures rely on inductive biases rather than explicit constraints on the scale or Fourier spectrum of the representation.

Other methods use coordinate networks for multiscale representation rather than efficient optimization. For instance, scale-aware positional encodings are trained at all scales simultaneously so that different output scales can be queried at inference time [10]. Band-limited coordinate networks (BACON) extend MFNs to control the network bandwidth, but require training on multiple output scales simultaneously for multiscale representation at inference time [8]. Our work is inspired by BACON, but significantly modifies the architecture, initialization scheme, and training techniques to make efficient multiscale optimization techniques compatible with coarse-to-fine reconstruction.

**Cryo-EM Reconstruction.** While coordinate networks have previously been used for cryo-EM reconstruction [5, 4, 6], they differ from our method in that they explicitly represent signals in the Fourier domain. Our goal is to demonstrate coarse-to-fine reconstruction methods with coordinate networks in the real domain, and we demonstrate this for cryo-EM reconstruction. The proposed method may also prove promising for advanced cryo-EM methods that resolve intrinsic motion of structures since spatial motion can be directly modeled in the primal domain [53].

## 3  Background

Coordinate neural networks have emerged as effective tools for approximating complex spatial data (e.g., see [1, 2, 7, 8, 10]). In the case of images, for example, they provide a mapping from continuous positions on the image plane to RGB values. Mapping 3D coordinates to volumetric density allows modeling 3D geometry. In particular, these networks have been shown to be readily trained to accurately approximate low-dimensional, complex signals in a memory-efficient way. Among this broad family of network architectures, in this paper we focus on Multiplicative Filter Networks (MFNs) [7] such as BACON [8], as they provide explicit control over the Fourier spectra of the function approximations.

**Multiplicative Filter Networks.** Most coordinate-based networks, like SIREN [2] and Random Fourier Features [3], use an MLP architecture consisting of the successive composition of linear

transformations and element-wise non-linearities. In contrast, Multiplicative Filter Networks (MFNs) use a Hadamard product (i.e., element-wise multiplication) instead of composition [7]. Concretely, in an $L$-layer MFN, the $d_{in}$ dimensional input coordinates, $\mathbf{x} \in \mathbb{R}^{d_{in}}$, are transformed using $L + 1$ non-linear filter modules, denoted by $g^{(i)}(\cdot) : \mathbb{R}^{d_{in}} \to \mathbb{R}^{d_h}$, $i = 0, 1, \ldots, L$, and $d_h$ is the hidden layer dimension. In [7], either *sinusoidal* or *Gabor* filters are used for this transformation. The sinusoidal case is given by $\mathbf{g}^{(i)}(\mathbf{x}) = \sin(\boldsymbol{\omega}^{(i)}\mathbf{x} + \boldsymbol{\phi}^{(i)})$, where $\boldsymbol{\omega}^{(i)} \in \mathbb{R}^{d_h \times 3}$ and $\boldsymbol{\phi}^{(i)} \in \mathbb{R}^{d_h}$ are referred to as frequencies and phases. At layer $i$, the intermediate representation of previous layer, $\mathbf{z}^{(i-1)}$, after being linearly transformed, is multiplied by the non-linear filter of input $\mathbf{g}^{(i)}(\mathbf{x})$. Formally,

$$
\begin{aligned}
\mathbf{z}^{(0)} &= \mathbf{g}^{(0)}(\mathbf{x}), \\
\mathbf{z}^{(i)} &= \mathbf{g}^{(i)}(\mathbf{x}) \odot \left(\mathbf{W}^{(i)}\mathbf{z}^{(i-1)} + \mathbf{b}^{(i)}\right), \quad i = 1, \ldots, L,
\end{aligned}
\tag{1}
$$

The behaviour of MFNs can be understood by analyzing the frequency of intermediate layers. Using the trigonometric identity

$$
\sin(a)\sin(b) = \frac{1}{2}\sin(a + b - \pi/2) + \frac{1}{2}\sin(a - b + \pi/2),
\tag{2}
$$

closed-form expressions for intermediate representations can be derived [7, 8]. Dropping the bias terms $\mathbf{b}^{(i)}$ for simplicity, the individual components of the intermediate representation $\mathbf{z}^{(i)}$ is equal to a weighted sum of an exponential number of sine terms [7]:

$$
\mathbf{z}_{n_i}^{(i)} = \sum_{\substack{\mathbf{n}=(n_0,\cdots,n_{i-1},n_i) \\ \mathbf{s}=(s_1,\cdots,s_i)}} \overline{\alpha}^{(i)}(n) \sin\left(\overline{\boldsymbol{\omega}}^{(i)}(\mathbf{n},\mathbf{s})\,\mathbf{x} + \overline{\phi}^{(i)}(\mathbf{n},\mathbf{s})\right)
\tag{3}
$$

where $\mathbf{n} = (n_0, \cdots, n_{i-1}, n_i)$ is a tuple of indices with $n_j \in \{1, \ldots, d_h\}$ and $s_j \in \{-1, 1\}$. Each term in the summation comprises an amplitude, frequency and phase shift, given by

$$
\overline{\alpha}^{(i)}(\mathbf{n}) = \frac{1}{2^i}\prod_{l=1}^{i}\mathbf{W}_{n_l,n_{l-1}}^{(l)}, \qquad
\begin{cases}
\overline{\boldsymbol{\omega}}^{(i)}(\mathbf{n},\mathbf{s}) &= \boldsymbol{\omega}_{n_0}^{(0)} + \sum_{l=1}^{i} s_l \boldsymbol{\omega}_{n_l}^{(l)} \\
\overline{\phi}^{(i)}(\mathbf{n},\mathbf{s}) &= \phi_{n_0}^{(0)} + \sum_{l=1}^{i} s_l(\phi_{n_l}^{(l)} - \frac{\pi}{2}).
\end{cases}
\tag{4}
$$

**BACON.** This analysis of the frequency content of an MFN shows that the bandwidth of $\mathbf{z}^{(i)}(\mathbf{x})$ is the sum of the input bandwidths. This was leveraged in BACON [8] to band-limit the representation of each individual layer. Additionally, an output was created for each layer to produce intermediate reconstructions; i.e.,

$$
\mathbf{y}^{(i)} = \mathbf{W}_{\text{out}}^{(i)}\mathbf{z}^{(i)} + \mathbf{b}_{\text{out}}^{(i)}.
\tag{5}
$$

where $\mathbf{W}_{\text{out}}^{(i)} \in \mathbb{R}^{d_{\text{out}} \times d_h}$ and $\mathbf{b}_{\text{out}}^{(i)} \in \mathbb{R}^{d_{\text{out}}}$. However, to encourage $\mathbf{y}^{(i)}$ to preserve its signal representation at scale $i$, one requires extra scale-specific losses during training, without which, subsequent training will corrupt the lower resolution representation.

**Motivating Example.** Although BACON does provide control over spectral bandwidth, when adopted in the context of a coarse-to-fine optimization procedure, it fails to preserve and carry the learned coarse-scale representations when one moves to the next finer-scale stage of the optimization. Consequently, at each round, one must re-optimize the entire representations at all previous scales, hindering efficient coarse-to-fine optimization. As an example, in Fig. 3 we examine the representation learned by BACON within a coarse-to-fine training strategy, where successive output layers are trained to fit an image at finer scales. We apply a staged training procedure such that a loss is applied to outputs at each scale in separate rounds of optimization. Unfortunately, when fitting finer scales BACON completely forgets the learned representations of coarse scale outputs from previous optimization rounds (see supplement). For a fairer comparison, given any round, we keep all scale-specific losses, but let only the linear output layers for other scales to be updated. Still, the results of BACON (Fig. 3, top row) show that the representations at coarser scales yet become corrupted while training at finer scales. In fact, reconstructions at different scales are highly coupled in BACON, since new layers can result in new frequencies anywhere within the band limit. In the following, we modify the architecture and its initialization and training scheme (Fig. 1) to make coordinate networks applicable to coarse-to-fine multiscale reconstruction.

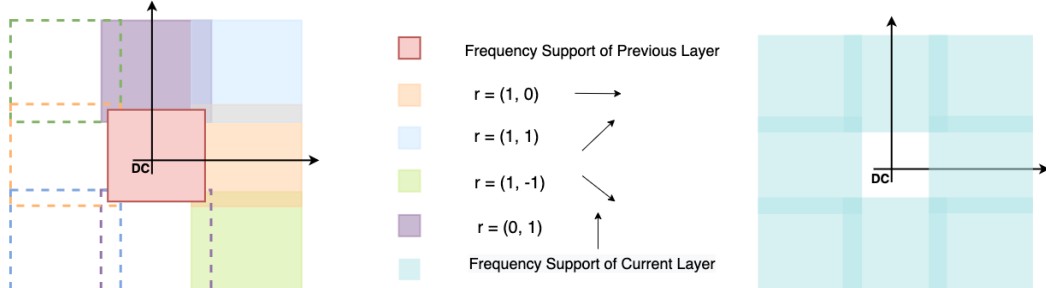

Figure 2: Illustration of the initialization scheme. (Left) The central red square depicts the base spectrum comprising frequencies of the previous layer $\overline{\boldsymbol{\varpi}}^{(i-1)}(\mathbf{n}, \mathbf{s})$. Other coloured regions depict copies of the base spectrum, shifted versions by $\lambda_2 B_{i-1}$ in four directions with $\lambda_2 \approx 2$. The copies are slightly larger due to the perturbation by the random $\mathbf{v}_j^{(i)}$. Dashed regions are naturally introduced because of the sign factors $s_i = \pm 1$. (Right) The new frequency spectrum of $\overline{\boldsymbol{\varpi}}^{(i)}(\mathbf{n}, \mathbf{s})$ is the union of the shaded regions. The initialization controls the overlap with the spectrum of the previous layer.

## 4  Residual Multiplicative Filter Networks

From Eq. 4, we can express frequencies in layer $i$ in terms of frequencies in previous layers with the recursion

$$\overline{\boldsymbol{\varpi}}^{(i)}(\mathbf{n}, \mathbf{s}) = \overline{\boldsymbol{\varpi}}^{(i-1)}(\tilde{\mathbf{n}}, \tilde{\mathbf{s}}) + s_i \boldsymbol{\omega}_{n_i}^{(i)} , \tag{6}$$

where $\tilde{\mathbf{n}}$ and $\tilde{\mathbf{s}}$ are formed from $\mathbf{n}$ and $\mathbf{s}$ by removing last entry, e.g., $\tilde{\mathbf{n}} = (n_0, \cdots, n_{i-1}) = \mathbf{n}_{<i}$. Thus $\mathbf{z}_{n_i}^{(i)}$ contains frequencies from the previous layer, shifted by $\boldsymbol{\omega}_{n_i}^{(i)}$ and $-\boldsymbol{\omega}_{n_i}^{(i)}$, where $\boldsymbol{\omega}_{n_i}^{(i)}$ is the frequency of hidden unit $n_i$ in layer $i$. More specifically, assuming that frequencies in the previous layer, i.e., $\overline{\boldsymbol{\varpi}}^{(i-1)}(\tilde{\mathbf{n}}, \tilde{\mathbf{s}})$, are band-limited to $[-B_{i-1}, +B_{i-1}]$, those in $\mathbf{z}^{(i)}$ lie in the union of several of its shifted clones. From this perspective, the frequencies at layer $i$, $\boldsymbol{\omega}^{(i)}$, characterize two important concepts: 1) the growth in bandwidth from the previous layer, and 2) the extent of overlap between the previous layer frequency spectrum and each new individual clone. For instance, given entries of $\boldsymbol{\omega}^{(i)}$ in $[-\delta, +\delta]$, the range of frequencies represented in $\mathbf{z}^{(i)}$ expands to $[-B_{i-1} - \delta, B_{i-1} + \delta]$. Also, if a specific $\boldsymbol{\omega}_{n_i}^{(i)}$ is set to a relatively large value of $2B_{i-1}$ this results in a clone with frequency support in the range $[B_{i-1}, 3B_{i-1}]$, and it avoids overlap with the original bandwidth $[-B_{i-1}, B_{i-1}]$.

### 4.1  Frequency Initialization

Motivated by this interpretation of $\boldsymbol{\omega}^{(i)}$, we propose a new initialization scheme which explicitly controls the overlap and expansion of frequencies in successive layers. For simplicity, we describe this initialization for the 2D case, though it can be extended to higher dimensions straightforwardly. First, we follow a procedure similar to BACON [8] to initialize and fix frequencies of filters which allows limiting outputs to specified bandwidths. Then, given the band-limited support $[-B_{i-1}, +B_{i-1}]$ of frequencies in layer $i - 1$, we initialize the 2D vectors $\boldsymbol{\omega}_j^{(i)}$ as follows:

$$\boldsymbol{\omega}_j^{(i)} = \lambda_2 B_{i-1} \mathbf{r}_j^{(i)} + \mathbf{v}_j^{(i)} ,$$
$$\mathbf{r}_j^{(i)} \sim U(\mathcal{R}), \qquad \mathbf{v}_j^{(i)} \sim U(-\lambda_1 B_{i-1}, \lambda_1 B_{i-1})^2 \tag{7}$$

where $\mathcal{R} = \{(0, 1), (1, 0), (1, 1), (1, -1)\}$, and $U$ denotes either continuous or discrete uniform distributions. Here, $\boldsymbol{\omega}_j^{(i)}$ is composed of two terms. The first term mainly specifies the direction $(r_j^{(i)})$ and the proportional amount $(\lambda_2 B_{i-1})$ by which frequencies of layer $i - 1$ will be shifted. The second term is a relatively small random perturbation to break the symmetry. An intuitive illustration of these operations is provided in Fig. 2. In 2D, there are 8 different directions to expand, half of which are redundant since $s_i = \pm 1$ entails two shifts in opposite directions $\pm(r_{j,1}, r_{j,2})$. Thus we only consider 4 possible values $R$. Using this initialization $\mathbf{z}^{(i)}$ is thus band-limited to $\overline{\boldsymbol{\varpi}}^{(i)} \in [-(1 + \lambda_1 + \lambda_2)B_{i-1}, (1 + \lambda_1 + \lambda_2)B_{i-1}]$.

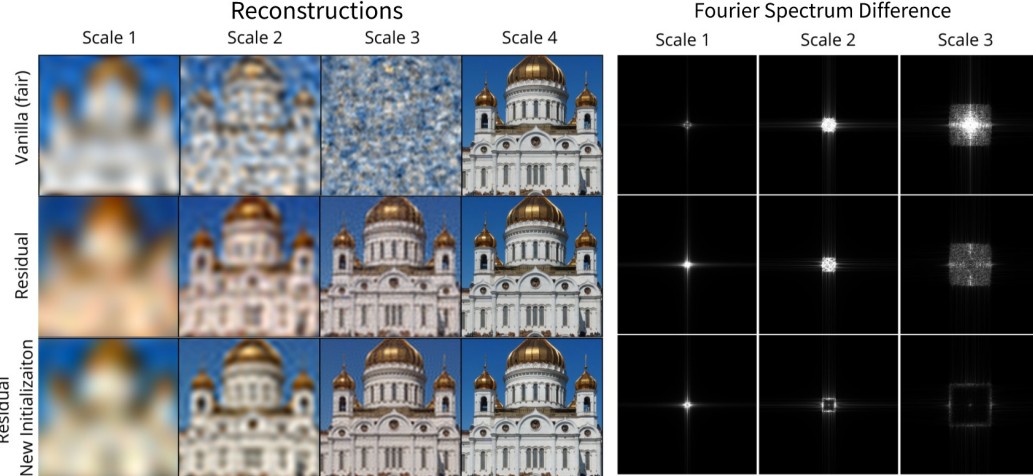

Figure 3: Coarse-to-fine image fitting experiments. We compare fair BACON (Vanilla), a residual MFN with the BACON initialization scheme, and a residual MFN with the proposed initialization. The proposed method performs best at retaining image quality at each scale during the staged training regiment. We also compare the difference between the Fourier spectra at each output scale immediately after the training stage and at the end of training (Right). In the proposed method, residual connections allow information at each scale to be re-used while other methods overwrite lower frequency information during coarse-to-fine training.

## 4.2 Residual Connections

Consider setting $\lambda_2 = 2 + \lambda_1$ during initialization. In this case, by design, the frequency spectrum for sine terms in layer $i$ would thus not overlap at all with that of layer $i - 1$; in at least one direction, the frequencies would either end up to left or right of interval $[-B_{i-1}, B_{i-1}]$. Thus the output layer would include no sine terms with frequencies in the central interval $[-B_{i-1}, B_{i-1}]$. To obtain the output of layer $i$, we sum it with the output of layer $i - 1$. That is,

$$\mathbf{y}^{(i)} = \mathbf{y}^{(i-1)} + (\mathbf{W}_{\text{out}}^{(i)} \mathbf{z}^{(i)} + \mathbf{b}_{\text{out}}^{(i)}). \tag{8}$$

This can be interpreted as a residual connection that ensures we keep an identical copy of frequencies from the previous layer; i.e., $\mathbf{y}^{(i-1)}$ will fill the gap in the frequency space. In practice, rather than the setting $\lambda_2 = 2 + \lambda_1$, one can use other less strict combinations which allow for different amounts of overlap and expansion. We found empirically setting $\lambda_2 = 2, \lambda_1 = 0.3$ to be effective.

## 5 2D Image Fitting

We first evaluate each of the proposed modifications to band-limited networks in the context of 2D image fitting. This task is useful for didactic purposes, allowing one to clearly see the benefits of skip connections and our new initialization scheme in the context of coarse-to-fine image reconstruction. We adopt a staged training procedure and fit a series of band-limited coordinate networks to natural images provided by [1]. The dataset consists of 32 RGB images, each of size $512 \times 512$. We first downsample the images by a factor of 2 to obtain images of size $256 \times 256$. These images are fit during training using a Mean Squared Error (MSE) training objective, applied to the output at the scale corresponding to the stage of training. As previously described, for the fairest comparison, the "Vanilla" BACON baseline is optimized with losses at all scales, though, at the stage of a given scale, only the linear output layers of other scales can be updated. Still, this keeps the optimization of the internal representations staged. All networks use a four-layer architecture and are trained for $10,000$ iterations, divided into chunks of $500, 500, 1000, 8000$ for optimizing output scales $\{1, 2, 3, 4\}$, respectively. The frequencies for all methods are initialized to have the same band-limit for consistency. All methods were implemented in PyTorch [54] and trained on a single NVIDIA GeForce RTX 2080.

Table 1: Coarse-to-fine image fitting results. We show the mean absolute difference of Fourier spectrum of outputs at each scale immediately after optimization and after the entire training regiment and peak signal-to-noise ratio for the image generalization task. Residual connections and the new initialization help to preserve the spectrum during training, especially for the second and third scales. The final reconstruction achieves competitive generalization results in terms of PSNR metrics.

| Method | Mean Abs. Difference | | | PSNR (dB) |
|---|---|---|---|---|
| | Scale 1 | Scale 2 | Scale 3 | Scale 4 |
| SAPE [9] | - | - | - | $27.02 \pm 3.29$ |
| BACON [8] | - | - | - | $28.39 \pm 3.66$ |
| (Staged) BACON (fair) | $\mathbf{1.61 \pm 0.45}$ | $7.36 \pm 1.84$ | $17.97 \pm 4.27$ | $\mathbf{28.46 \pm 3.61}$ |
| (Staged + Res) BACON | $4.33 \pm 0.97$ | $5.16 \pm 1.22$ | $9.23 \pm 2.73$ | $27.90 \pm 3.27$ |
| (Staged + Res + Init) BACON | $3.60 \pm 0.95$ | $\mathbf{3.42 \pm 0.85}$ | $\mathbf{4.32 \pm 1.35}$ | $28.33 \pm 4.42$ |

Figure 3 visualizes the outputs at each scale for BACON, the residual MFN with the band-limited BACON initialization, and the residual MFN with our proposed initialization. Since the coarse-to-fine training scheme can potentially disturb previously trained outputs when applying the loss at the finer scales, we visualize the differences between the Fourier spectra at each output scale after the corresponding training iterations and after the full training regiment (Fig. 3, right). The proposed method shows the least disturbance after training. For the entire image dataset, we report the Fourier spectrum mean absolute difference and standard deviation in Table 1.

Finally, we examine all networks on the image generalization task by querying them on a finer grid of $512 \times 512$ and computing the Peak Signal-to-Noise Ratio (PSNR) between reconstructed signal and the original ground truth. The average PSNR values over all images is summarized in the last column of Table 1. We also benchmark SAPE [9] as a baseline for coarse-to-fine reconstruction, on the image generalization task. SAPE gradually exposes higher frequencies as a function of time and space, while providing only a single reconstruction at a time, so only their full-scale reconstruction can be evaluated for comparison. While our approach improves interpretability of the reconstructions, by enabling reuse of previous learned representations, it achieves empirical performance competitive with baselines. Additional experiments on a textual dataset are provided in Appendix A.

## 6 Cryo-EM 3D Reconstruction

Single particle cryo-EM is an increasingly popular experimental technique to recover the 3D structure of macromolecular complexes, such as proteins and viruses. Cryo-EM has gained widespread attention in the last decade with pioneering advances in hardware and data processing techniques. This has enabled atomic or near-atomic resolution reconstruction of challenging structures [55]. Multiscale, coarse-to-fine reconstructions has long played a critical role in cryo-EM estimation, e.g., [20], making it an ideal testbed for our proposed methods.

**Image Formation Model.** Based on a simplified model [56], the protein structure is represented as a function $V : \mathbb{R}^3 \to \mathbb{R}_{\geq 0}$, which maps points in 3D space to non-negative real values, expressing the Coulomb potential induced by the atoms. Images $\{I_i\}_{i=1}^n$ are captured with a transmission electron microscope which are 2D orthogonal projections of randomly oriented potential maps. Each image $I_i$ has a corresponding latent pose (orientation) $R_i \in SO(3)$ representing the 3D rotation of the molecule in the image. In practice, to ensure sufficient contrast, the microscope is defocused when images are captured, resulting in a point-spread function (PSF) $g_i$. To minimize radiation damage to delicate biological molecules, samples are exposed to low dosages of electrons resulting notoriously noisy images. Taken together, the image formation model can be expressed as

$$I_i^*(x,y) = g_i \star \int_{\mathbb{R}} V(R_i^T \mathbf{x}) \, dz + \epsilon, \quad \mathbf{x} = (x,y,z)^T \tag{9}$$

where the projection is, by convention, assumed to be along the $z$-direction after rotation, $\star$ corresponds to convolution and $\epsilon$ is Gaussian noise [14, 4]. The PSF can be estimated in a preprocessing step and is assumed here to be known; for more details see [57]. (A more detailed formulation for cryoEM reconstruction is given in Appendix B.)

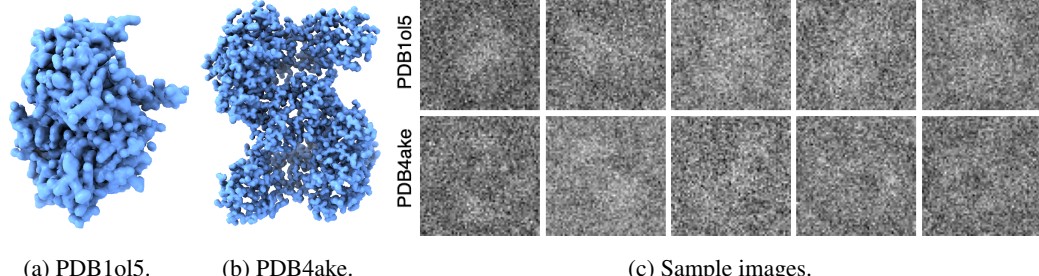

(a) PDB1ol5.    (b) PDB4ake.          (c) Sample images.

Figure 4: Ground truth potential maps and cryo-EM measurements. (Left) Potential maps generated using the corresponding atomic models. (Right) Noisy example images from a synthetically generated particle stack after applying the PSF and adding white Gaussian noise.

**3D Reconstruction Problem.** The goal of a cryo-EM experiment is to estimate the unknown 3D potential map of the biomolecule present in the observed images $\{I_i\}_{i=1}^n$. This task has some intrinsic challenges. First, the images are extremely noisy (see Fig. 4c). Second, the ground-truth poses are typically unknown, or only a rough estimate is available, so it is not possible to perform 3D reconstruction using the forward model of Eq. 9 directly. Previous work [6, 4, 19] estimate pose parameters during reconstruction with various techniques including amortized inference, variational bounds or direct marginalization. Here we use a stochastic gradient-based method, which optimizes the representation of the 3D structure $V$, parameterized by our proposed residual MFN, together with the unknown pose variables $\{R_i\}_{i=1}^n$. The loss function is the squared error between the observed image $I_i$ and that generated by the model, i.e.,

$$\mathcal{L} = \sum_{x,y} \left( I_i(x,y) - I_i^*(x,y) \right)^2 \tag{10}$$

where $I_i^*(x,y)$ is given above in Eq. 9. See Appendix D for more detail on the pose parameterization and optimization.

In practice, naively optimizing for both the poses and potential map suffers from poor local minima. To avoid this, coarse-to-fine reconstruction approaches are commonly employed. At early stages, low frequency content of the map is estimated as the SNR is higher so estimation is easier, which then allows for more reliable subsequent pose estimation as higher frequencies are included. Previous approaches [4, 14] employ this technique, called *Frequency Marching* [20], as they model the map in the Fourier domain where individual frequencies are readily accessible. In contrast, we address the reconstruction problem in a coarse-to-fine manner while expressing the potential map in real space.

## 6.1 Experiments

We evaluated our approach on two synthetic datasets which allow us to directly evaluate against ground truth structures. The datasets are based on atomic models of 1OL5 (https://www.rcsb.org/structure/1ol5) and 4AKE (https://www.rcsb.org/structure/4ake) from Protein Data Bank (PDB). We first convert atomic models to $64 \times 64 \times 64$ discrete potential maps, with voxel sizes 1 Å and 1.2 Å respectively, using *pdb2mrc.py* module from EMAN2 [58]. Then we simulate the image formation model as in Equation 9, at $50,000$ rotations uniformly sampled on $SO(3)$. To generate each image, we rotate the map $V$, resample it on the original grid, project it along canonical z-direction and apply the PSF. Zero-mean white Gaussian noise is added to these simulated clean images produce a realistic SNR of 10%. The structures and sample images are visualized in Fig. 4.

**Experimental Settings.** We use a 3-layer network with 128 hidden units to represent the 3D structure. The network outputs the structure in 3 hierarchical scales and, accordingly, the training process is divided into 3 stages, one for each scale. We set the hyperparameters as $\lambda_1 = 0.3, \lambda_2 = 2.0$ and train the network for a total of 100 epochs. For PDB1ol5, we spend respectively 15, 15, and 70 epochs optimizing the first to the third scales. For PDB4ake, we find a better reconstruction if we dedicate more epochs on the first and second scales, namely 25 epochs on each of the first and second scales. When computing the loss, images are band-limited to the corresponding scale using a Gaussian filter whose scale parameter is set to match the bandwidth of the current stage

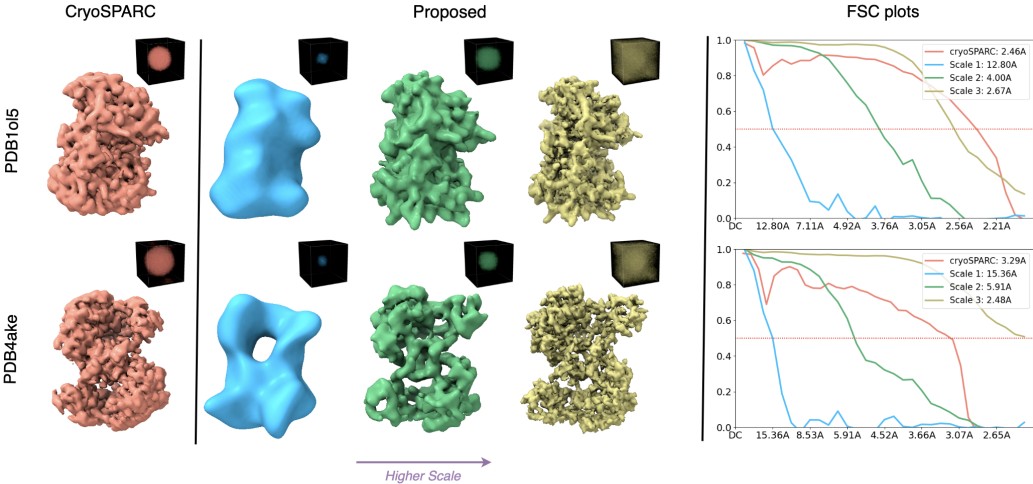

Figure 5: (Left) CryoSPARC reconstructions. (Middle) Our reconstruction at multiple scales. (Right) Fourier Shell Correlation plot measuring correlation in Fourier spectrum between reconstructions and ground-truth structure. Threshold 0.5 is the criterion to compute the resolutions in Angstroms.

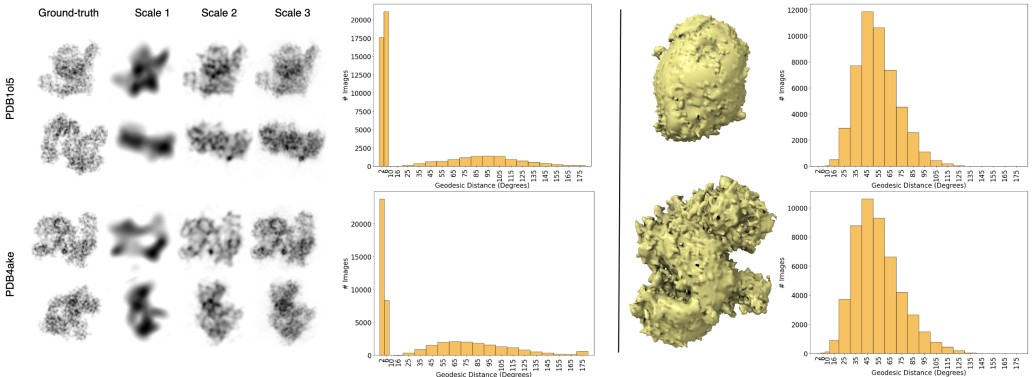

Figure 6: (Left) Histogram of geodesic distances between estimated and ground-truth poses. Some example reconstructed images are illustrated as well. The first row for each dataset contains a particle with the estimated pose within at most 4 degrees away from ground-truth while the second row corresponds to another with totally wrong pose estimate. (Right) Getting stuck at local minima when optimizing the full-scale reconstruction from scratch (not coarse-to-fine). Although, the overall shape is well captured, no significant improvement is achieved during several iterations and poses are stuck at local minima, mostly concentrated around 45 degrees away ground-truth.

scale. Mini-batches of images are used to update poses and weights of the network representing the structure. For each mini-batch, we alternate between optimizing the structure and poses for $5$ and $20$ iterations, respectively. We use Adam optimizer [59] with a learning rate $0.001$ for the network parameters and a higher learning rate of $0.01$ for pose parameters. Each pose parameter is initialized to random value uniformly distribution over a geodesic distance of $45$ to $90$ degrees from the ground truth. This can be considered as a very crude initial estimate for poses. In Appendix E we consider even coarser initial pose values, showing that they achieve competitive qualitative and quantitative performance to those here. Experiments are implemented in PyTorch [54], models are trained on 4 NVIDIA TITAN Xp GPUs, and ChimeraX [60] is used to visualize the 3D maps.

**Results.** Qualitative comparison of reconstructions with those obtained by homogeneous refinement of cryoSPARC (as the baseline) are illustrated in Fig. 5. In the middle, we observe that, from low to high scales, increasingly more structural features are resolved. The low-scale reconstructions capture only the global and low-resolution features, whereas the full-scale ones contain fine-grained details. Our approach is able to well-resolve some peripheral parts of the structure, especially for the more

complex structure of PDB4ake. Also there are some structural features learned by our reconstruction yet totally missed by cryoSPARC. All in all, our final reconstruction visually looks competitive or better when compared with the baseline.

As shown on the right side of Fig. 5, to quantitatively evaluate reconstructions, we compute the Fourier Shell Correlations (FSCs) [61] between the reconstructed map and the ground-truth. FSC is the gold-standard metric in cryo-EM area measuring the normalised cross-correlation coefficient between two 3D volumes in Fourier domain along shells of increasing radius (see Appendix C for more background on FSC). We measure FSC between ground-truth and reconstructions at all scales, following the convention of the threshold $0.5$ to compute final resolution, as ground-truth is known *a priori* in our experiments. We also evaluate how well the poses are optimized and converge to the ground-truth. To this end, the histogram showing number of images with various geodesic distances between estimated and ground-truth poses are visualized as in Fig. 6. For both datasets, there is a concentration of images on left hand side corresponding to slight error in estimation of pose parameters. Thus, for most of the images, predicted pose parameters are within few degrees of the corresponding ground truth, verifying that poses are optimized well along with the reconstruction. Furthermore, per dataset, we select two sample images, one with almost the correct pose estimate, and the other with one that is far from ground truth. Then we demonstrate their reconstruction at all scales as in Fig. 6. Finally, we conduct an experiment in which we supervise the network only at the full-scale reconstruction throughout the entire training. The output structures are depicted in the right panel of Fig. 6, realizing a situation where poses are trapped in local minima, and we thus observe no substantial improvement in the structure for many iterations.

## 7 Conclusions, Limitations and Future Work

In this paper, we build on Band-limited Coordinate Network (BACON) and propose a new architecture and training strategy providing fine-grained control over spectral characteristics of network outputs, yielding a multiscale representation suitable for coarse-to-fine reconstruction. The combination of skip connections and a consistent initialization scheme allows the proposed network to ensure lower levels of the network represent the coarser scales of the signal which are then reused at finer scales. We empirically show that these modifications enable coordinate networks to fit natural images in coarse-to-fine fashion. They are also shown to be effective for cryo-EM reconstruction.

**Limitations.** We find that the proposed multi-scale optimization procedure is effective in avoiding local minima in the Cryo-EM reconstruction problem. We have yet to explore whether residual MFNs are also effective on other problems with local minima due to the existence of nuisance variables. Although beyond the scope of this paper, it is also possible that minor modifications of the architectural and hyper-parameter settings would improve the robustness of the optimization; e.g., increasing the number of layers for more fine-grained scales, or a more diverse inspection of $\lambda$ hyperparameters. Moreover, the cryo-EM experiments in this paper used synthetic data, to enable more thorough the evaluation of the method. Real datasets are likely to require the use of other bases as well, e.g., *Gabor*, which are better able to represent the spatial variability of resolution in cryo-EM reconstructions. We have also focused on homogeneous and rigid structures. An advantages of our approach is that the efficient multiscale representation of the signal in the real domain opens promising doors for modeling flexible structures, a major open problem in cryo-EM [4, 53].

**Societal Impact.** Cryo-EM for macromolecular structure determination is one of the foremost exciting areas in molecular biology with enormous social impact. Cryo-EM tools were key to determining the structure of the SARS-COV2 spike protein, the discovery of its pre-fusion conformation, and the assessment of potential medical countermeasures. Advances in this field enable the discovery and exploration of large protein complexes and viruses more generally [62]. On the other hand, we strongly condemn any usage of our proposed coordinate networks for generating malicious representations, improperly modifying signals, or spreading misinformation.

## Acknowledgments and Disclosure of Funding

We thank Ali Punjani for numerous valuable discussions and for use of the cryoSPARC software package. This research was supported in part by the Province of Ontario, the Government of Canada, through NSERC, CIFAR, and the Canada First Research Excellence Fund for the Vision: Science to Applications (VISTA) programme, and by companies sponsoring the Vector Institute.

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
