# Appendix

# Residual Multiplicative Filter Networks
# for Multiscale Reconstruction

## A    2D Image Fitting.

We provide additional natural image fitting results in Figs. 8 and 9. We also include the case of training "unfair" vanilla BACON in a staged fashion which, unlike the proposed fair variant, does not continue to update the linear output layer during the subsequent optimization stages of each scale. For the generalization task, we visualize the final reconstructions of SAPE [9], BACON [8], and our proposed method and compare it with ground truth as in Fig. 10.

We further investigate the proposed modifications on a new dataset of 32 RGB text images, each of size $512 \times 512$, provided by [1]. Similar to natural images, we first obtain downsampled $256 \times 256$ images. For all networks, we use the same architecture as above, in the main body of the paper. The models are fit to images using Mean Squared Error (MSE) as the objective loss. The only difference is that we also visualize learned reconstructions for the unfair case of staged training of vanilla BACON. We keep all the experimental settings same, including the number of iterations dedicated to each stage and the learning rates. Further, the frequencies for all networks are initialized to have the same band-limit.

In Fig. 11, the outputs at each scale for both unfair and fair staged BACON, the residual MFN with the band-limited BACON initialization, and the residual MFN with our proposed initialization, are visualized. The differences between the Fourier spectra at each output scale after the corresponding training iterations and after the full training regiment are illustrated too (Fig. 11, right). We provide additional examples of fitting text images in Fig. 12, comparing reconstruction at coarser scales of two and three. We also report the Fourier spectra mean absolute difference and standard deviation in Table 2 for the text dataset. Finally all networks are examined on the same image generalization task and the Peak Signal-to-Noise Ratio (PSNR) between reconstructed signal and the original ground truth is computed. We include SAPE [9] as a baseline for coarse-to-fine reconstruction. Average PSNR values over all images is summarized in the last column of Table 2. Similarly, visualization of the final reconstruction on this task is provided in Fig. 13.

Table 2: Coarse-to-fine image fitting results on Text dataset [1]. The mean absolute difference of Fourier spectra of outputs at each scale immediately after optimization and after the entire training regiment is reported. With skip connections and the new initialization, we observe better conservation of spectra on scales two and three. Also the peak signal-to-noise ratio (PSNR) on the image generalization task is reported indicating that the final reconstruction is on par with baselines.

| Method | Mean Abs. Difference | | | PSNR (dB) |
|---|---|---|---|---|
| | Scale 1 | Scale 2 | Scale 3 | Scale 4 |
| SAPE [9] | - | - | - | $30.99 \pm 2.32$ |
| BACON [8] | - | - | - | $31.54 \pm 2.31$ |
| (Staged) BACON (fair) | $\mathbf{1.06 \pm 0.29}$ | $3.57 \pm 1.35$ | $13.79 \pm 4.60$ | $31.57 \pm 2.29$ |
| (Staged + Res) BACON | $3.22 \pm 1.06$ | $3.77 \pm 1.04$ | $8.11 \pm 2.60$ | $31.17 \pm 2.77$ |
| (Staged + Res + Init) BACON | $3.01 \pm 0.51$ | $\mathbf{2.99 \pm 0.99}$ | $\mathbf{3.56 \pm 1.16}$ | $\mathbf{32.04 \pm 2.34}$ |

## B    Cryo-EM

Single particle cryo-EM is a revolutionary imaging technique used by structural biologists to discover the 3D structure of macromolecular complexes, including proteins and viruses. Recovering 3D structure is a major step towards understanding how these miniature biological machines perform actions in our body cells. During sample preparation for a cryo-EM experiment, one first obtains a purified sample in solution, containing many instances of a specimen. This sample is then plunged into a cryogenic liquid, like liquid ethane, which freezes the molecules at random orientations. Once frozen in vitreous ice, the sample is loaded into a transmission electron microscope and exposed to

parallel electron beams which are recorded underneath by special detectors. The majority of these electrons pass through the structure without interaction, causing shot noise. The rest are scattered either in an elastic or inelastic fashion. Elastic scattering positively contributes to image formation by introducing a relatively weak phase contrast while the inelastic scattering damages the structure leading to garbage particles.

The image formation in cryo-EM can be well-approximated using the weak-phase object model [56]. This model assumes the 3D structure is a density map $V : \mathbb{R}^3 \to \mathbb{R}_{\geq 0}$ represented under a canonical orientation. Ideally, without any corruption, as depicted in Fig. 14, one can formalize clean images $\{I_i^*\}_{i=1}^n$ as 2D orthogonal projections of the map under random orientations $R_i \in SO(3)$,

$$I_i^*(x, y) = (\mathcal{P} V_{R_i})(x, y) ,$$
$$\mathcal{P} V_{R_i} = \int_{\mathbb{R}} V_{R_i}(x, y, z) dz , \tag{11}$$
$$V_{R_i}(x, y, z) = V(R_i^T (x, y, z)^T) .$$

Here, $\mathcal{P}$ is a linear operator performing the orthogonal projection and $V_{R_i}$ corresponds to a structure rotated by $R_i$. However, in practice, images are intentionally captured under defocus in order to improve the contrast. This is modeled by convolving the clean images with an image-specific point-spread functions (PSF) $g_i$. We express all sources of noise with an additional term and, taken together, the observed image can be formally described as

$$I_i(x, y) = (g_i \star I_i^*)(x, y) + \epsilon(x, y). \tag{12}$$

Here, $\star$ denotes convolution, $\epsilon$ is the additive noise, and $V_{R_i}(x, y)$ corresponds to clean (noiseless) projection of the structure rotated by $R_i$, as in Fig. 14. Statistically, as several factors contribute to the noise, it is common practice to assume $\epsilon$ is white (or colored) Gaussian noise.

## C   Fourier Shell Correlation (FSC) and Resolution

Fourier shell Correlation (FSC) is the standard way to measure the resolution of a 3D density map. In a nutshell is is the correlation between the Fourier coefficients of two aligned maps, as a function of frequency [61]. Let $S_r$ be a spherical shell with radius $r$ centered at the origin of the Fourier domain (i.e., containing 3D frequencies whose wavelengths are close to $1/r$). Then, given two maps in the Fourier domain, $F_1$ and $F_2$, FSC$(r)$ is defined as the normalized cross-correlation between the Fourier coefficients of $F_1$ and $F_2$ within $S_r$; i.e.,

$$\text{FSC}(r) = \frac{\sum_{k \in S_r} F_1(k) F_2^*(k)}{\sqrt{\sum_{k \in S_r} |F_1(k)|^2 \sum_{k \in S_r} |F_2(k)|^2}} . \tag{13}$$

One can show that FSC$(r)$ is real-valued, and ranges from -1 to 1. Once correlations are computed for all shells, we plot FSC as a function of $r$. In practice, one defines $D/2$ shells for maps with $D^3$ voxels. Typpically, one of $F_1$ or $F_2$ is the grount truth and the other is an estimated map. Or the dataset is randomly partitioned into two halves, used to independently estimate the two maps $F_1$ or $F_2$. At small $r$ (i.e., low frequencies), where observed images have strong signal, maps are higher quality and FSC is close to 1. With higher frequencies the maps become noisier and FSC therefore decreases (see the right panel of Fig. 5). When a reconstruction is compared to a ground-truth map, one usually defines the resolution of the estimated map as the $r$ for which FSC$(r) = 0.5$. Under mild assumptions this corresponds to a SNR of 1 [63].

It is worth noting that FSC also plays a key role in a more advanced version of frequency marching. In experimental settings, as the true map is not available, FSC is computed between two independent reconstructions. During frequency marching, as we increase the frequency support, the FSC curve is used to cross-validate the refinement to each reconstruction. In fact, due to high frequency noise in the data, refinements are likely to overfit independently, yielding inconsistencies in the subsequent high frequency regime. This is often observed as a sudden drop in the FSC curve. And when this is detected, we adaptively limit the frequency support [64].

In our work, we achieve high quality reconstructions without this adaptive regularization. On the other hand, we only have few scales for computational efficiency and this adaptive version is justified when more scales are dedicated. We leave further investigations of this issue to the future work.

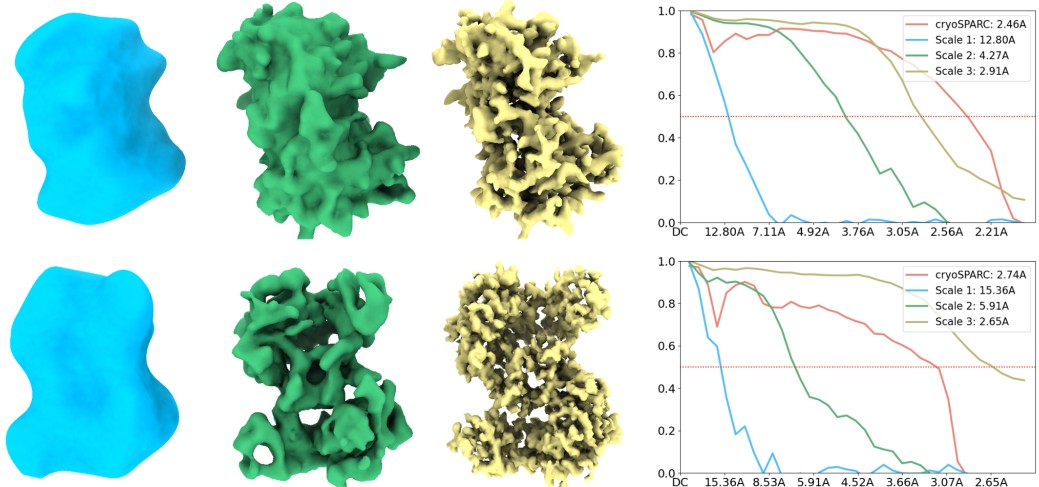

Figure 7: Results on coarser initialization of poses. (Left) Our reconstruction at multiple scales. (Right) Fourier Shell Correlation plot measuring correlation in Fourier spectrum between reconstructions and ground-truth structure. Threshold $0.5$ is the criterion to compute the resolutions in Angstroms.

## D  Pose Optimization

Here, we provide the details of learning pose parameters, describing the 3D orientation of the molecule in an observed given particle image. Let us denote current estimate of pose parameters by $\hat{R}_i \in SO(3)$. By applying a small perturbation $\delta R_i \in SO(3)$, one can obtain a new estimate for poses by multiplication $(\delta R_i)\hat{R}_i$. Since, $SO(3)$ is a group closed under multiplication, the perturbed pose $(\delta R_i)\hat{R}_i$ remains a valid rotation. To find a better estimate, we iteratively find the optimal perturbation, and accordingly update pose estimates at each iteration. To parameterize the perturbation $\delta R_i$, similar to [21], we adopt the exponential map which is a mapping from the tangent space to the points on the $SO(3)$ manifold. One way to realize the exponential map is through axis-angle representation $(\theta_i, \mathbf{u}_i)$, where $\theta_i$ is a scalar angle between zero and $\pi$ and $\mathbf{u}_i \in \mathbb{R}^3$ is a unit vector. One can simply use Rodrigues formula to get back to the matrix form

$$\delta R_i = I + \sin(\theta_i)[\mathbf{u}_i]_\times + (1 - \cos(\theta_i))[\mathbf{u}_i]_\times^2 \ , \tag{14}$$

where $[\mathbf{u}_i]_\times$ denotes the skew-symmetric $3 \times 3$ matrix of $\mathbf{u}_i$. We rely on gradient-based optimization to solve for optimal perturbation parameters $(\theta_i, \mathbf{u}_i)$

$$(\hat{\theta}_i, \hat{\mathbf{u}}_i) = \underset{(\theta_i, \mathbf{u}_i)}{\arg\min} \mathcal{L}\left((\theta_i, \mathbf{u}_i)|V, I_i\right), \tag{15}$$

where the image-specific loss is conditioned on the fixed potential map $V$ and the corresponding image $I_i$. We iteratively compute gradients with respect to the pose parameters and then update them. Note that, after such an update, the resulting pose angle may no longer lie in the valid interval $[0, \pi]$, or the new axis $\hat{\mathbf{u}}_i$ may no longer have unit length. For this reason we re-establish these constraints by truncating angles back to $[0, \pi]$, and normalizing $\mathbf{u}_i$ to be a unit vector.

## E  Coarser Initialization of Poses

We conduct further experiments where poses are initialized to be uniformly distribution with angles within $[0, \pi]$ of the ground-truth. In this more challenging coarser initialization of pose, we dedicate more epochs to optimize the first and second scales, namely 30 epochs each. Qualitatively, we obtain reconstructions on par with those in the main paper. In terms of our FSC resolution metric, we achieve full-scale resolutions of 2.91A and 2.65A for PDB1ol5 and PDB4ake, respectively, as shown in 7.

# F  $\lambda$ **hyperparameters**

The $\lambda$ hyperparameters control the growth and disjointness of frequency support in the residual MFN (see Sec. 4.1). Our first intuition was to avoid settings that create gaps in coverage of the Fourier domain, i.e., when $\lambda_2 > 2 + \lambda_1$. Beyond this constraint, we investigated a range of hyper-parameter settings in our experiments. In all cases, the network fits the signal well; but we found that for $\lambda_2 = 2$, $\lambda_1 = 0.3$, the model yields a lower test loss (based on held out data for image fitting experiments). For cryoEM experiments (3D regression) we found the same hyperparameters perform well, based on FSC with ground truth 3D density. We therefore used the same hyperparameters for cryo-EM experiments with 2D images, without further hyper-parameter search.

# G  **Videos**

For further visualization of cryo-EM experimental results, we use Chimera [60] to create a set of videos, provided under *videos* folder, showing the results obtained by our method. In videos *pdb1ol5_all_scales.mp4* and *pdb4ake_all_scales.mp4*, we visualize final 3D structures learned at different scales, respectively for pdb1ol5 and pdb4ake. Reconstructions are rotated in the visualization to show the resolved structural features from several views.

We further visualize the coarse-to-fine evolution of the structure for each dataset during the entire training in videos *pdb1ol5_evolution.mp4* and *pdb4ake_evolution.mp4*. In both cases, we start from a random structure. We then obtain a coarse reconstruction and later stages capture more fine-grained (i.e. high-resolution) details of the estimated structure.

Finally, we use Chimera to fit the available atomic model of pdb4ake into the corresponding final reconstruction. As shown in video *model_fit.mp4*, various secondary structures, such as $\alpha$-helices, are well aligned with the reconstructed 3D density map.

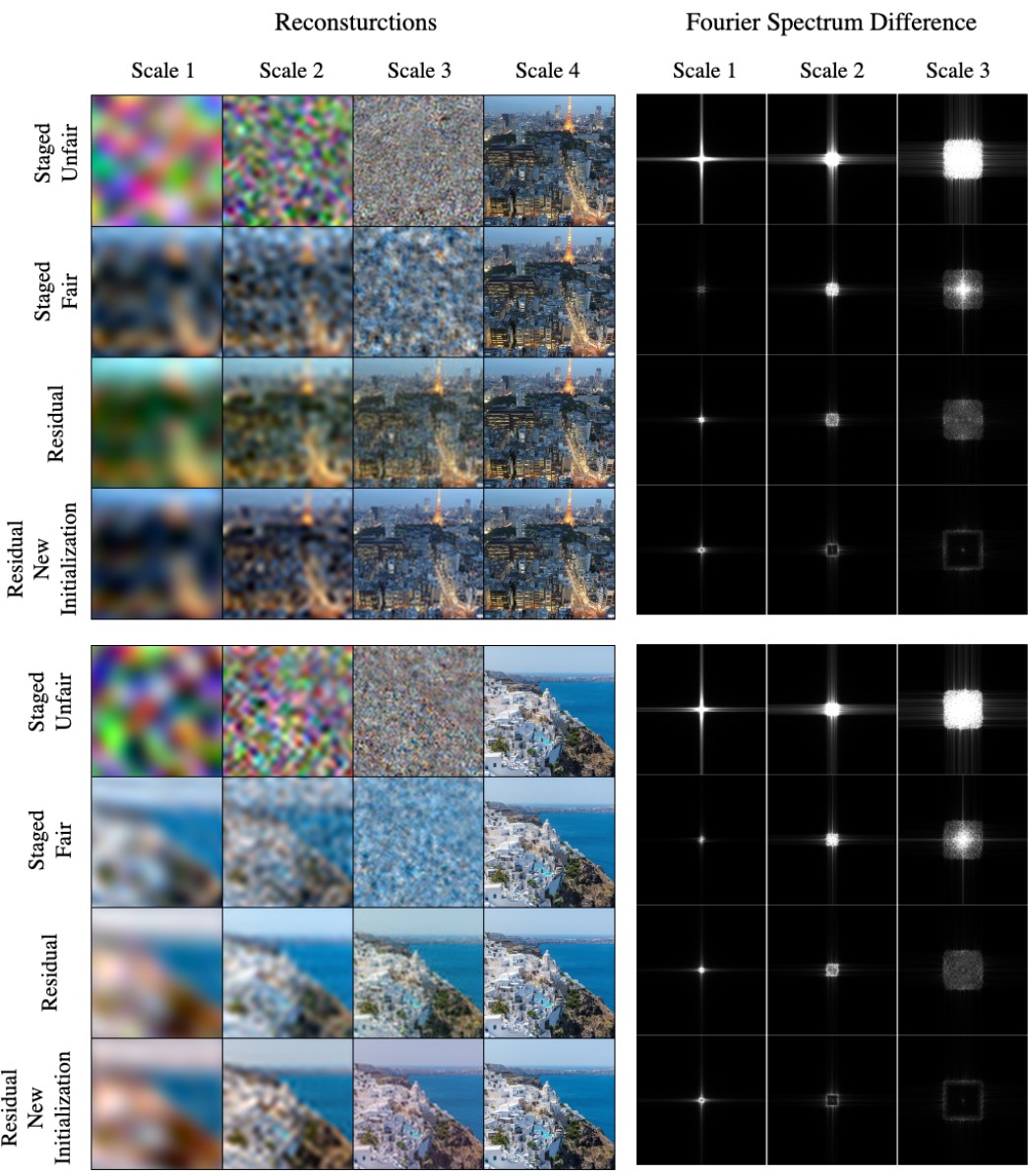

Figure 8: Coarse-to-fine fitting experiment on two example Natural images. (Left) Reconstructions depicted at all scales. (Right) Disruption to Fourier spectra of coarser scales caused by later training stages.

| Staged Unfair | Staged Fair | Staged Residual | Staged Residual New Initialization | Staged Unfair | Staged Fair | Staged Residual | Staged Residual New Initialization |

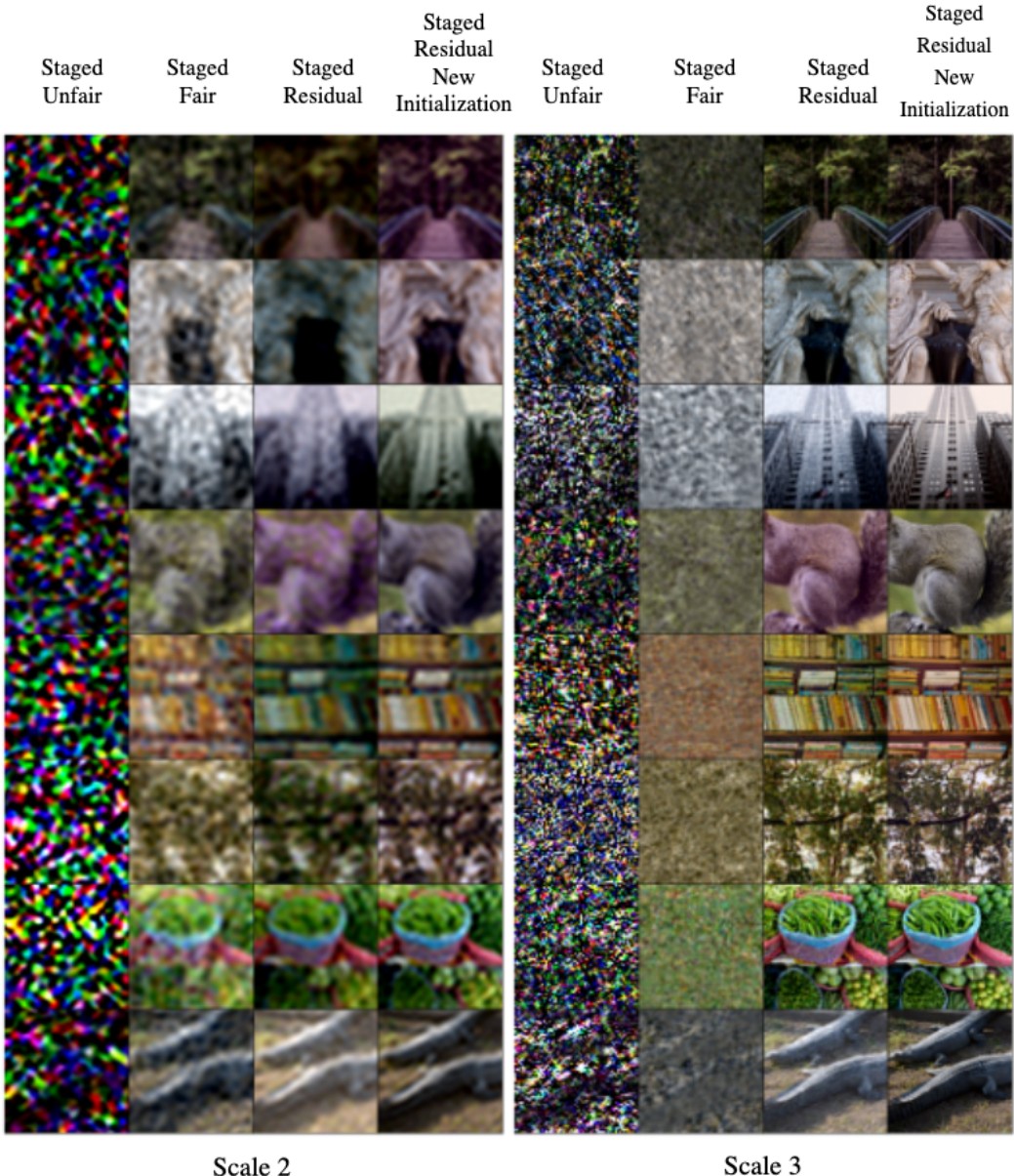

Scale 2                                   Scale 3

Figure 9: Coarse-to-fine fitting experiment on more examples of Natural images. Qualitative comparison of reconstructions at scales 2 (Left) and 3 (Right) for different methods imply that both residual connections and the new initialization noticeably improve the preservation of spectra at lower scales, especially scale 3.

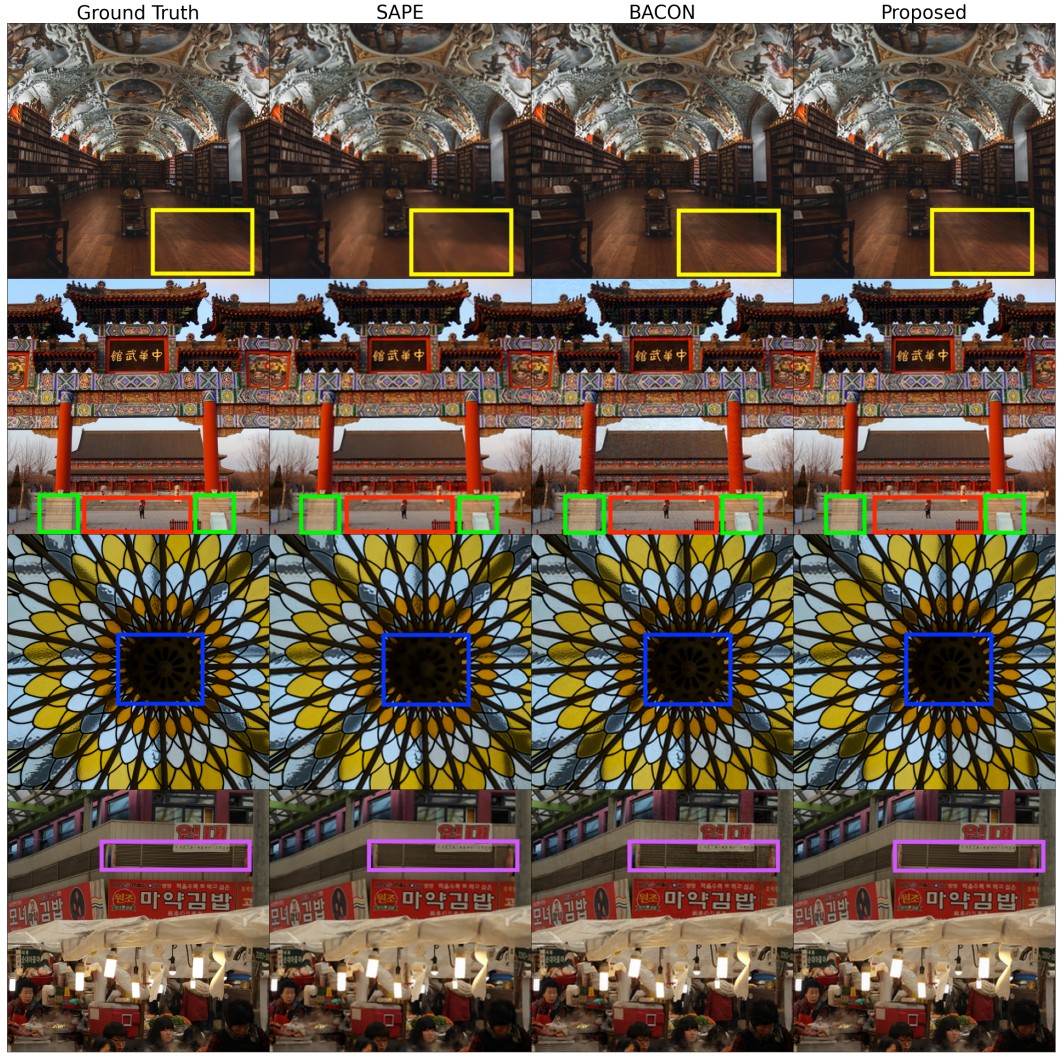

Figure 10: Examples on Image Generalization on Natural dataset. Some areas are highlighted with colored boxes showing regions where differences among reconstructions or from the ground truth can be observed.

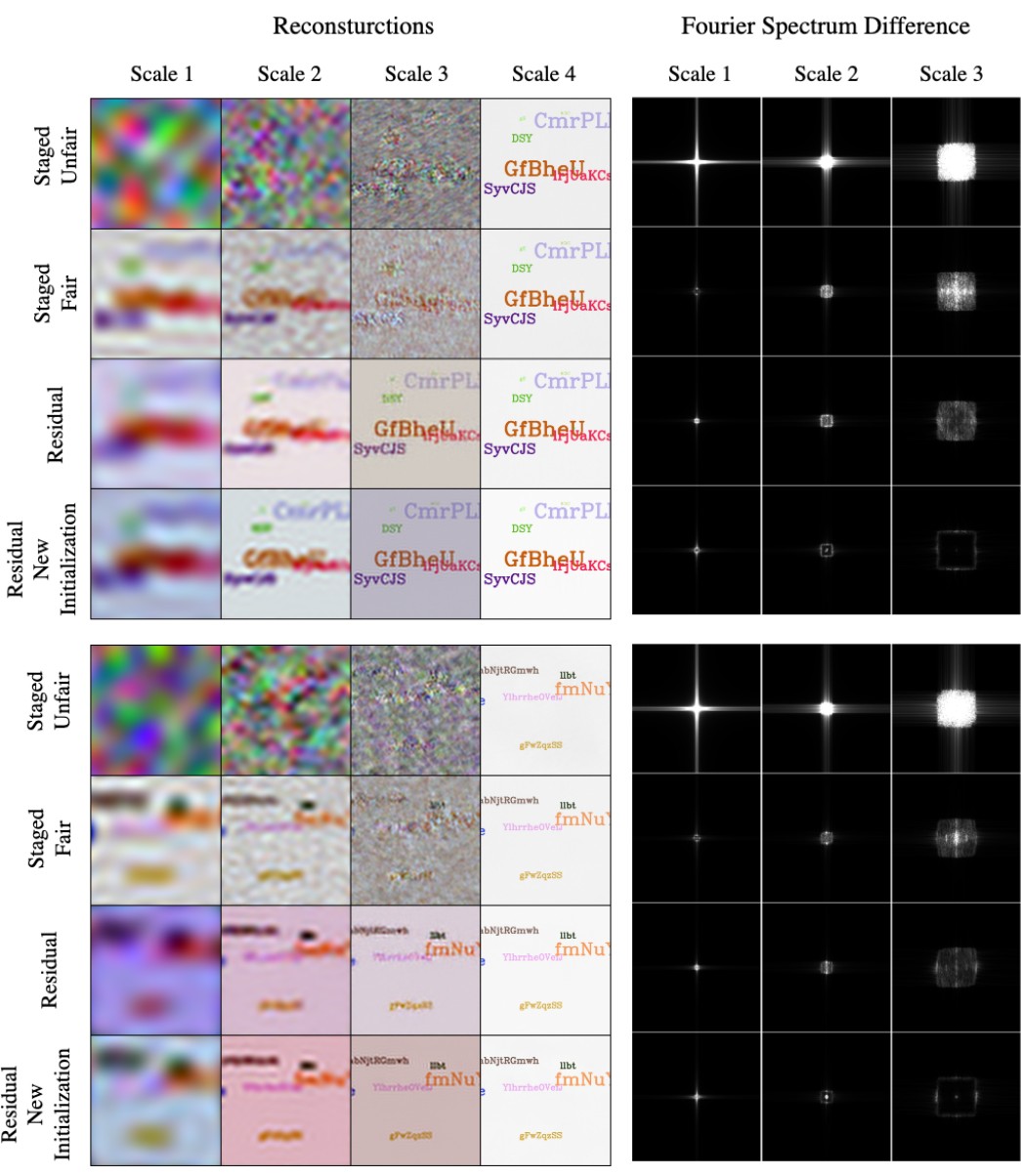

Figure 11: Coarse-to-fine fitting experiments on two example Text images. (Left) Reconstructions depicted at all scales. (Right) Disruption to Fourier spectra of coarser scales caused by later training stages.

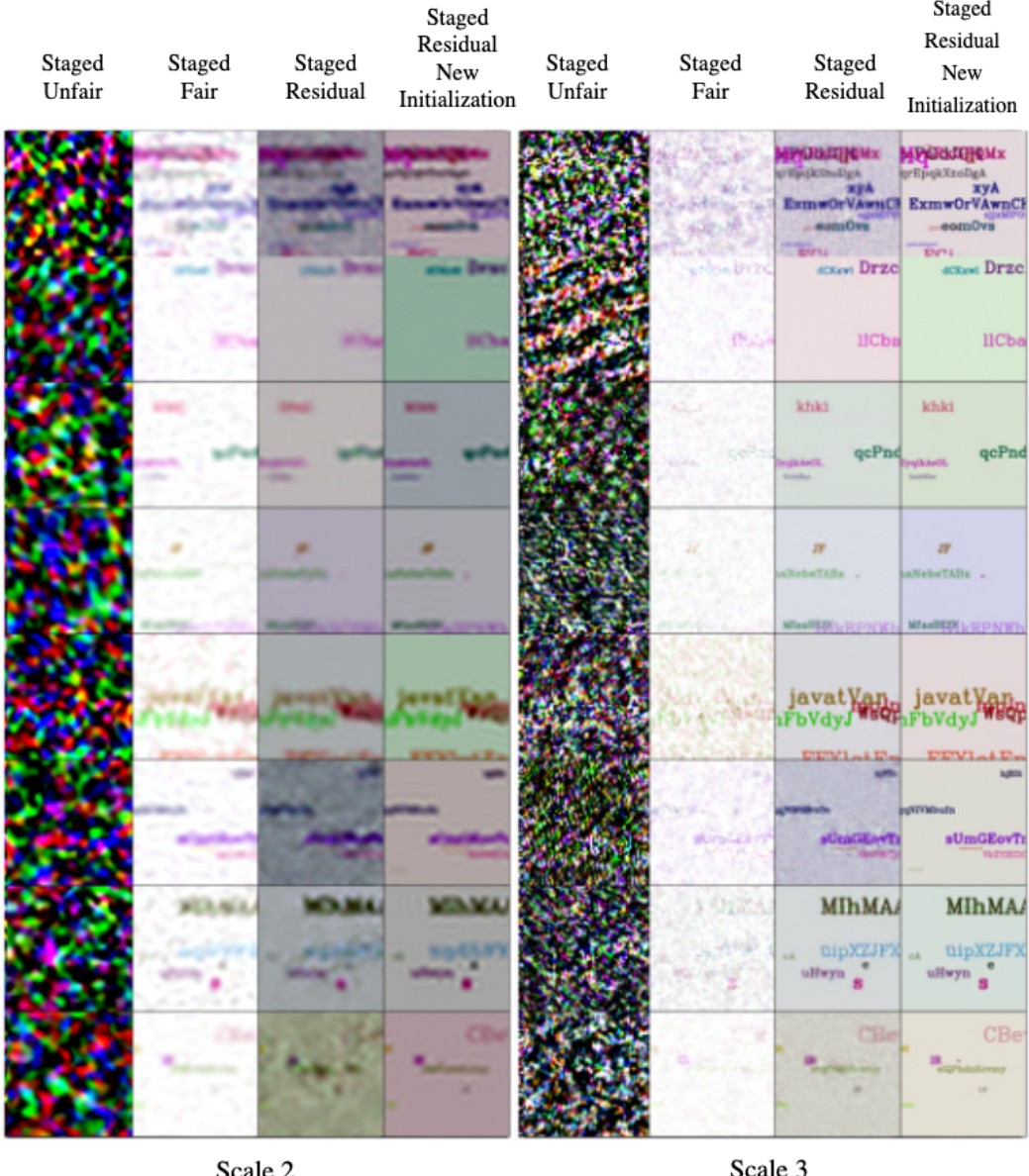

| Staged Unfair | Staged Fair | Staged Residual | Staged Residual New Initialization | Staged Unfair | Staged Fair | Staged Residual | Staged Residual New Initialization |

Scale 2                                           Scale 3

Figure 12: Coarse-to-fine fitting experiment on more examples of Text images. Qualitative comparison of reconstructions at scales 2 (Left) and 3 (Right) for different methods imply that both residual connections and the new initialization noticeably improve the preservation of spectra at lower scales, especially scale 3.

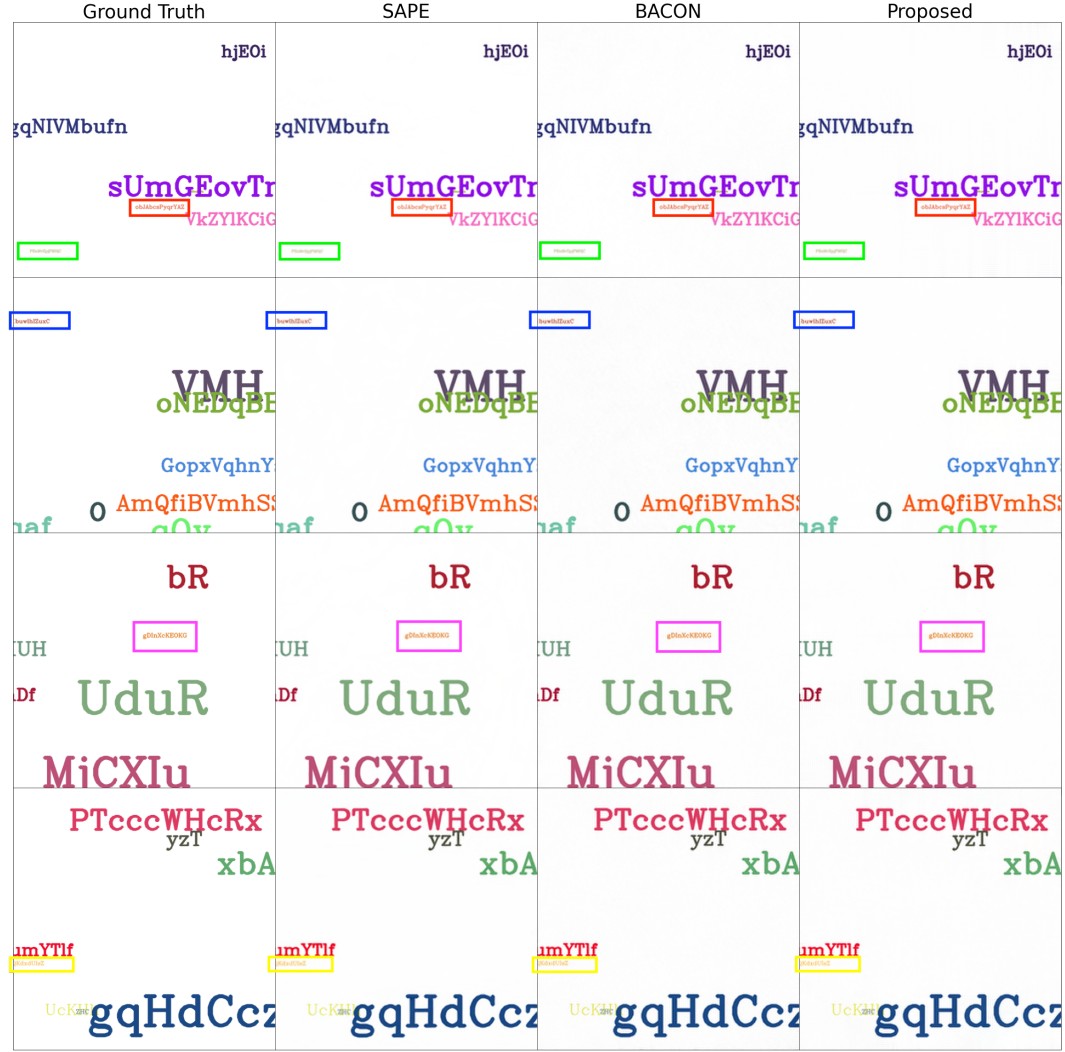

Figure 13: Examples on Image Generalization on Text dataset. Some areas are highlighted with colored boxes showing regions where differences among reconstructions or from the ground truth can be observed.

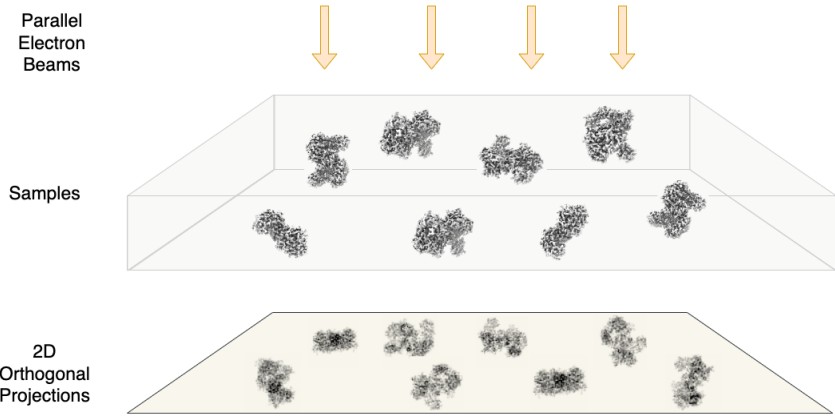

Figure 14: Ideal image formation model generating clean 2D orthogonal projections of samples.