# OpenReview forum: "Residual Multiplicative Filter Networks for Multiscale Reconstruction"
_NeurIPS.cc/2022/Conference — NeurIPS 2022 Accept_

### Official Review · Reviewer_Vy34 · 2022-07-11

**Rating:** 5
**Confidence:** 4
**Soundness:** 2 fair
**Presentation:** 3 good
**Contribution:** 2 fair

**Summary:**

The authors treat the problem of coarse-to-fine optimization for solving inverse problems, which are particularly important for 3D reconstruction. They note that the existing state-of-the-art coordinate networks used to solve these problems are ill-suited for multiscale optimization since they can forget low frequency details learned during earlier stages in the training scheme. The authors propose residual multiplicative filter networks which (1) introduce skip connections between layers with different output scales so that coarse features learned early in the optimization process are explicitly not forgotten and (2) use an initialization scheme that allows for control over the frequency spectra learned by each layer, meaning that the network can avoid learning redundant frequency information in finer-scale layers. Several experiments are presented for 2D and 3D reconstruction problems where the proposed method is shown to be superior to existing approached in capturing details at several scales.

**Questions:**

Related to my first point in the weaknesses section, it would be useful if the authors could provide a description of why the multi-scale metrics (e.g. as shown in the FSC plots in Fig. 5) matter in the context of reconstruction problems. My current understanding is that these metrics are provided to demonstrate that the proposed method has stable training during multi-scale optimization and that it successively adds fine-grained detail due to the innovations presented in the paper. However, I am not sure if I am supposed to interpret these metrics as useful for actually solving the reconstruction or inverse problem. My intuition is that as long as the final (i.e. the last scale during training) output is consistently useful, then the intermediate training qualities do not matter nearly as much.

**Limitations:**

The authors provide a small list of limitations in the second paragraph of the conclusion. However, the section reads as a justification for some design and experimental choices rather than a discussion about the shortcomings of the proposed method. I would suggest that the authors make a more explicit limitations (sub)section.

**Strengths And Weaknesses:**

STRENGTHS:

- I like the intuitive breakdown of the intermediate representations as a sum and shift of frequency sprectra from the previous layers. This provides a straightforward motivation for the proposed method.

- The visual representation in Figure 6 gives a clear and concrete example of why coarse-to-fine optimization is important for 3D reconstruction problems.

- Detailed background is given on 3D molecular reconstruction problems, which is useful for the larger ML community that may not be familiar with such domains.


WEAKNESSES:

- I do not understand the benefits of having the outputs at each scale "shift" very little (e.g. as in the results in Table 1). I would think that as long as the final output PSNR is satisfactory, then the internal shift in frequency repsresentation does not matter. Looking at Table 1, it seems that even though fair BACON has quite a bit of shift in its frequency representations at each scale before and after optimizing the other scales, the output still has good PSNR.

- I believe that a more thorough background and introduction to coordinate-based networks in general could be given. While some details are given (e.g. that the networks take 2D or 3D coordinates and output some value like RGB), it was not immediately clear to me what the purpose of these networks (e.g. compressing a high-dimensional signal) was compared to other models (e.g. generative models).

---

> ### Author Response · Authors · 2022-08-02
> **Response to Reviewer Vy34**
>
> Thank you for the comments and questions in the review. Below we summarize and address the concerns that were raised in the review:
>
> **“...a more thorough background and introduction to coordinate-based networks…”**
> We agree that more background would be very helpful. In the revised version of the paper, we will add more background on coordinate networks to the introduction to make the work more accessible.
>
> **“...as long as the final output PSNR is satisfactory, then the internal shift in frequency representation does not matter...why the multi-scale metrics (e.g. as shown in the FSC plots in Fig. 5) matter in the context of reconstruction problems…the intermediate training qualities do not matter nearly as much.”**
>
> We agree that for the problem of image fitting, the goal is often to obtain a high fidelity output at the final layer. With that said, we included this experiment for didactic reasons, like previous work [1-5], and show that, for other methods, fitting finer scales of the signal ruins the reconstruction already learned at previous stages.
>
> For the Cryo-EM experiments, on the other hand, it is crucial that accurate reconstructions be achieved at each scale in order that the pose estimates converge reliably. Such coarse-to-fine optimization can only be achieved if successive layers in the network work collaboratively with previous layers to refine the reconstruction.
>
> As a further example, consider trying to use BACON for coarse-to-fine optimization in Cryo-EM. After optimizing a coarse reconstruction of pose and structure at the first output layer, we find that the optimized low frequency components are not carried over to the subsequent output layer, but become corrupted. Thus optimization at finer scales cannot proceed directly from the initial coarse reconstruction and must waste time correcting information at lower scales that had already been determined. Our proposed residual connections and initialization scheme avoids this because the optimized low frequency signal components are carried over exactly to the subsequent output layers.
>
> **“...suggest that the authors make a more explicit limitations (sub)section.”**
> We will expand the limitations section in the paper. In the new version, we will discuss limitations related to hyperparameter selection (such as determining bandwidths and number of epochs per training stage), discrete scale outputs, and other potential architectural limitations. These are all interesting areas for future research on the less-explored multi-scale MFN models.
>
> **References.** \
> [1] ​​R. Fathony, A.K. Sahu, D. Willmott, J.Z. Kolter, Multiplicative filter networks, in: International Conference on Learning Representations, 2020. \
> [2] D.B. Lindell, D. Van Veen, J.J. Park, G. Wetzstein, Bacon: Band-limited coordinate networks for multiscale scene representation, in: Proceedings of the IEEE/CVF Conference on Computer Vision and Pattern Recognition, 2022: pp. 16252–16262. \
> [3] A. Hertz, O. Perel, R. Giryes, O. Sorkine-Hornung, D. Cohen-Or, Sape: Spatially-adaptive progressive encoding for neural optimization, Advances in Neural Information Processing Systems. 34 (2021) 8820–8832. \
> [4] M. Tancik, P. Srinivasan, B. Mildenhall, S. Fridovich-Keil, N. Raghavan, U. Singhal, R. Ramamoorthi, J. Barron, R. Ng, Fourier features let networks learn high frequency functions in low dimensional domains, Advances in Neural Information Processing Systems. 33 (2020) 7537–7547. \
> [5] V. Sitzmann, J. Martel, A. Bergman, D. Lindell, G. Wetzstein, Implicit neural representations with periodic activation functions, Advances in Neural Information Processing Systems. 33 (2020) 7462–7473.

---

> > ### Comment · Reviewer_Vy34 · 2022-08-05
> > **Response to Author Rebuttals to Reviewer Vy34**
> >
> > > In the revised version of the paper, we will add more background on coordinate networks to the introduction to make the work more accessible.
> >
> > Thank you for this. I believe this will make the paper more self-contained and easy to understand for researchers who are outside of this sub-field.
> >
> > > We agree that for the problem of image fitting, the goal is often to obtain a high fidelity output at the final layer. With that said, we included this experiment for didactic reasons, like previous work [1-5], and show that, for other methods, fitting finer scales of the signal ruins the reconstruction already learned at previous stages....
> >
> > Thank you for providing references for this precedent. I now understand that these experiments are included mainly for presentation quality. I hope that the authors will include this explanation or a similar one in the main text for clarification purposes.
> >
> > > As a further example, consider trying to use BACON for coarse-to-fine optimization in Cryo-EM. After optimizing a coarse reconstruction of pose and structure at the first output layer, we find that the optimized low frequency components are not carried over to the subsequent output layer, but become corrupted. Thus optimization at finer scales cannot proceed directly from the initial coarse reconstruction and must waste time correcting information at lower scales that had already been determined. Our proposed residual connections and initialization scheme avoids this because the optimized low frequency signal components are carried over exactly to the subsequent output layers.
> >
> > Thank you for the clarification. I like the pointed example of 3D reconstruction to illustrate the utility of the proposed methods. As before, I hope the authors will include this point in a revision of the main text.
> >
> > > We will expand the limitations section in the paper. In the new version, we will discuss limitations related to hyperparameter selection (such as determining bandwidths and number of epochs per training stage), discrete scale outputs, and other potential architectural limitations. These are all interesting areas for future research on the less-explored multi-scale MFN models.
> >
> > I appreciate this expansion of the paper. The authors sufficiently address my concerns regarding the limitations of the present work. Overall, I think my concerns have been addressed to a satisfactory degree, and I will raise my review score for this paper.

---

### Official Review · Reviewer_9iUQ · 2022-07-11

**Rating:** 6
**Confidence:** 2
**Soundness:** 3 good
**Presentation:** 3 good
**Contribution:** 3 good

**Summary:**

The paper describes an extension of the recently proposed Multiplicative Filter Networks (MFN) that permits an independent parametrization of different levels of scale (Fourier-bandwidth) by introducing residual connections and a suitable initialization scheme.
MFNs are a simplification of positionally encoded networks / coordinate networks that have a special form of building the result as a linear combination of (potentially) exponentially many basis functions that are controlled non-linearly, achieved by multiplication with Fourier (or Garbor) basis functions in each "level" of the network (in the 2D/3D case considered here, the dimension of the output [function] space does not seem to be exponential though, bringing this closer to a classical linear representation) . The related BACON network extends this idea by providing initialization schemes that yield nested levels of scale (increasing resolution), but the networks are constructed in a way such that changes to higher level "fine" details invalidate the fit of lower layers.
The submission addresses this issue in a simple but effective way by adding short-cut connections that carry the lower-layer function-fits to the next-higher layer unaltered, and building-up a new detail representation on top (which, to my understanding, can show complex scale dependencies). In addition, a suitable initialization is proposed that creates nested levels of detail in the Fourier-spectrum covered (again, to my understanding, targeting the low-dimensional regime of maybe 2D or 3D coordinates).
Results for image fitting are improved over previous methods such as BACON. An example application to CryoEM reconstruction is presented that seems to outperform a well-known approach (characterized as SOTA).

The main contribution is a scale-separated version of MFN/BACON. The applications to CryoEM are also interesting but the rather brief evaluation with only a few example reconstructions move the methodological contribution more in focus.


**Questions:**

- Does it makes sense to compare against linear base-lines? If so, what results would you get (or expect)? If not, why is it more expressive (and would it still be if the scales were fully separated [if I understood this correctly])?
- Is the method scalable to larger dimensions?

**Limitations:**

If my understanding of the limitations in terms of dimensionality were correct (see discussion question above), it would be useful to discuss this more in the paper.
It might also be useful to mention the limitations of multi-scale optimization for avoiding local minima (in my experience with shape matching & reconstruction problems only very vaguely related to the CryoEM example, it helps somewhat but does not eliminate fundamental issues with bad local minima).

**Strengths And Weaknesses:**

To explain my view of strength and weaknesses, I would first discuss my understanding of the technical aspects a bit more in depth:

I understand the main goal of the paper as to decouple scales in an MFN-style coordinate network. To my understanding, this is already achieved by the short-cut connections, which yield a superposition of the low-frequency part, solely controlled by low-frequency parameters, with a high-frequency part that depends on both low- and high-frequency parameters. For progressive coarse-to-fine-optimization, this seems to be sufficient, while at the same time being conceptually and technically very easy to understand and implement. A suitable initialization makes sure that progressive parts of the spectrum are covered (by introducing corresponding control parameters); this scheme seems to work only in moderately high dimensions (as it needs to cover Fourier-coefficient space with 2^d or 3^d patches of coefficients). Consequently, it is not fully clear how the decoupling would affect higher-dimensional applications, but they are not the target of the paper (and a lot of previous work, where similar representations are used for example for reconstruction from 3d point clouds).

I think that the concern for scale separation is a valid goal, and results on image fitting in the paper at least indicate subtle advantage of the decoupled representation (although a strong explanation is missing; the case for CryoEM is clearer as multi-scale fitting is commonly used there). It would be useful to understand more clearly how important this is in an overparametrized regime, as multi-scale optimization is often used as a heuristic to counter bad local minima in non-overparametrized, non-convex optimization scenarios.

From a conceptual point of view, I do not fully understand the boundaries (as in representational capacities and ability to find solutions) between linear scale space approaches (mip-maps, wavelets), MFN-approaches (linear in $x$, non-linear in $\theta$), and the current proposal. If - hypothetically - the next-higher frequency band was fully separated by duplicating the required low-res coefficients, would the resulting method be substantially different from an ordinary hierarchy of Fourier-basis functions? How different (how much better) are results in comparison to a simple base-line (say, a linear wavelet or pyramid representation)? It would be useful to see a direct base-line comparison (or maybe arguments for why this is clearly inferior - I do not know this area too well).

Summarized as a list, I would see the following strength...
+ Simple and straightforward idea. Easy to implement.
+ Canonical extension of MFN/BACON (users of those methods probably want to know about this one).
+ Good results with minimal (extra) effort (Seems to yield SOTA on CryoEM)
+ Well written paper.

...and weaknesses:
- Experimental evaluation remains a bit anecdotal (I guess that the CryoEM fitting is very expensive to setup and conduct).
- Very simple, maybe incremental idea (could also be seen as a limitation of the contribution)
- Limited explanation of "why" and "how far" it works.
- Could have more comparisons against base-line

Overall, I would guess that the paper might be a simple but frequently employed addition to MFN-methods and thus potentially impactful (with many citations and applications). For this reason, I tend towards a positive recommendation.

---

> ### Author Response · Authors · 2022-08-02
> **Response to Reviewer 9iUQ**
>
> We appreciate your thoughtful feedback. In what follows, we address the main concerns and questions individually:
>
> **“...results on image fitting…indicate subtle advantage of the decoupled representation (although a strong explanation is missing…”**
> While our multiscale representation is not required for high-fidelity image fitting, we believe the example is useful pedagogically. We use image fitting to clearly explain the initialization scheme and how residual connections facilitate coarse-to-fine optimization. Finally, image fitting has become a standard baseline for evaluating representational effectiveness of coordinate-based networks [1-5], and we show that our architecture performs on par with other methods.
>
> **“...how important this is in an overparameterized regime…”**
> While overparameterization is effective for many problems in deep learning, we suspect that it would not be sufficient to avoid poor local minima in cryo-EM. In particular, the energy landscape for poses in cryoEM is low dimensional, and due to non-convexity and very low SNR, local minima are highly problematic. Thus, whether or not an over-parameterized network is used to represent the 3D density map, pose optimization remains prone to local minima. With that said, given (accurate) pose estimates, fitting an overparameterized MFN model may be effective in avoiding poor local minima in parameter space, thanks to more friendly loss landscapes [6].
>
> **“If - hypothetically - the next-higher frequency band was fully separated..., would the resulting method be substantially different from an ordinary hierarchy of Fourier-basis functions?”
> “...the boundaries (as in representational capacities and ability to find solutions) between linear scale space approaches (mip-maps, wavelets), MFN-approaches (linear in $x$, non-linear in $\theta$), and the current proposal…comparison to a simple base-line…”**
>
> A conventional Fourier based representation would use a basis comprising orthogonal sinusoids. As such, computation of the optimal representation is straightforward. While a residual MFN also provides a representation consisting of sinusoids, the corresponding frequencies do not necessarily lie on a regular grid. Thus, regardless of full-separation of frequency bands, the bases are not orthogonal over a finite domain. Moreover, a conventional Fourier based representation stores amplitudes and phases explicitly, whereas the forward pass in MFN computes the linear sum over sinusoids whose amplitudes and phases are implicitly represented by the network parameters.
>
> Coordinate-based networks thus offer significant advantages compared to conventional linear basis representations. For example, in terms of memory, using the residual MFN allows us to optimize a volume of $64^3 \sim 260K$ voxels with only $\sim 50K$ free variables. This disparity increases for higher resolution volumes: In the computer vision literature, coordinate-based networks typically have a memory footprint of a few MB vs several GB for recent voxel-based models (e.g., [7, 8] for $\sim512^3$ volumes). There are also potential benefits in terms of computational complexity. Consider representing an $N^3$ volume with a sinusoidal basis function (similar to MFNs). Given a basis of $N^3$ frequency components, generating an output at a single coordinate requires evaluating a sum over $O(N^3)$ multiplications. For MFNs, generating an output requires $O(D^2)$ multiplications, where $D$ is the dimensionality of the hidden layers. Note that, in general, with coordinate networks, we empirically need a moderate increase in width and/or depth to fit a signal at larger scale [9].
>
> While we believe approaches based on residual MFNs or linear basis functions could achieve similar reconstruction quality, residual MFNs offer a significantly reduced memory footprint and potentially reduced computational complexity. Moreover, by leveraging coordinate-based networks we pave the way for future work leveraging powerful data-driven priors tailored to these methods [10,11].
>
> **“...experimental evaluation remains a bit anecdotal…”**
> For cryoEM experiments, we provide quantitative and qualitative evidence in Fig. 5 that the proposed method achieves reconstructions competitive with the state of the art. We use the FSC curve, the gold-standard metric in the community, to numerically evaluate the reconstructions (visualized in the right panel of Fig. 5). We also follow the convention of reporting the resolution based on the frequency by which the FSC drops below 0.5, and obtain high resolutions of 2.67A and 2.48A for PDB1ol5 and PDB4ake, respectively. Moreover, the qualitative results on the left panel of Fig. 5 show that our method is capable of recovering fine-grained details. We provide additional animated visualizations in the supplementary material (model_fit.mp4), which clearly show that our final reconstruction for PDB4ake is well-aligned with the ground-truth atomic model.

---

> > ### Author Response · Authors · 2022-08-02
> > **Response to Reviewer 9iUQ (Cont.)**
> >
> > **“...maybe incremental idea…”**
> > While we build on previous work on MFN-style architectures, our work is the first to propose an interpretable, multiscale coordinate-based network compatible with coarse-to-fine optimization techniques. This enables more efficient convergence and allows us to use coordinate-based architectures to tackle challenging inverse problems (e.g., Cryo-EM reconstruction). We believe our method broadens the appeal of coordinate-based architectures, and will inspire follow-on work in Cryo-EM and other communities outside of machine learning.
> >
> > **“...Is the method scalable to larger dimensions?...”**
> > Coordinate-based networks have generally been applied to relatively low-dimensional signals. Our work also explores the common setting of reconstructing 2D and 3D signals. Extending these architectures to higher dimensions is a general problem that remains to be tackled in the literature. We suspect that MFN-like architectures may suffer a curse of dimensionality due to the need for explicit support across the Fourier domain; however, this question requires more investigation and is outside the direct scope of our work.
> >
> > **“...the limitations of multi-scale optimization for avoiding local minima… it helps somewhat but does not eliminate fundamental issues with bad local minima”**
> > In practice, we found that the proposed multi-scale optimization is effective in avoiding local minima for the Cryo-EM reconstruction problem. Still, it would be interesting to explore if our approach can successfully avoid local minima in other problem domains. While this is beyond the scope of the current project, it’s possible that minor architectural modifications could improve the robustness of our scheme in such settings, e.g., increasing the number of layers to achieve a more regimented coarse-to-fine optimization. Our work will spur follow-on work investigating the effectiveness of design choices in avoiding local minima by multi-scale optimization. We will include a discussion of this point in the paper.
> >
> > **References.** \
> > [1] ​​R. Fathony, A.K. Sahu, D. Willmott, J.Z. Kolter, Multiplicative filter networks, in: International Conference on Learning Representations, 2020. \
> > [2] D.B. Lindell, D. Van Veen, J.J. Park, G. Wetzstein, Bacon: Band-limited coordinate networks for multiscale scene representation, in: Proceedings of the IEEE/CVF Conference on Computer Vision and Pattern Recognition, 2022: pp. 16252–16262.
> > [3] A. Hertz, O. Perel, R. Giryes, O. Sorkine-Hornung, D. Cohen-Or, Sape: Spatially-adaptive progressive encoding for neural optimization, Advances in Neural Information Processing Systems. 34 (2021) 8820–8832.
> > [4] M. Tancik, P. Srinivasan, B. Mildenhall, S. Fridovich-Keil, N. Raghavan, U. Singhal, R. Ramamoorthi, J. Barron, R. Ng, Fourier features let networks learn high frequency functions in low dimensional domains, Advances in Neural Information Processing Systems. 33 (2020) 7537–7547. \
> > [5] V. Sitzmann, J. Martel, A. Bergman, D. Lindell, G. Wetzstein, Implicit neural representations with periodic activation functions, Advances in Neural Information Processing Systems. 33 (2020) 7462–7473. \
> > [6] H. Li, Z. Xu, G. Taylor, C. Studer, T. Goldstein, Visualizing the loss landscape of neural nets, Advances in Neural Information Processing Systems. 31 (2018). \
> > [7] A. Yu, S. Fridovich-Keil, M. Tancik, Q. Chen, B. Recht, A. Kanazawa, Plenoxels: Radiance fields without neural networks, ArXiv Preprint ArXiv:2112.05131. (2021). \
> > [8] B. Mildenhall, P.P. Srinivasan, M. Tancik, J.T. Barron, R. Ramamoorthi, R. Ng, Nerf: Representing scenes as neural radiance fields for view synthesis, in: European Conference on Computer Vision, Springer, 2020: pp. 405–421. \
> > [9] E. Dupont, H. Kim, S. Eslami, D. Rezende, D. Rosenbaum, From data to functa: Your data point is a function and you should treat it like one, ArXiv Preprint ArXiv:2201.12204. (2022). \
> > [10] D. Ha, A. Dai, Q.V. Le, Hypernetworks, ArXiv Preprint ArXiv:1609.09106. (2016). \
> > [11] C. Finn, P. Abbeel, S. Levine, Model-agnostic meta-learning for fast adaptation of deep networks, in: International Conference on Machine Learning, PMLR, 2017: pp. 1126–1135.

---

### Official Review · Reviewer_bz3B · 2022-07-13

**Rating:** 7
**Confidence:** 4
**Soundness:** 4 excellent
**Presentation:** 3 good
**Contribution:** 3 good

**Summary:**

This paper proposes two modifications to the recently introduced multiplicative filter networks (and their bandlimited variant BACON): 1) by introducing skip connections it enables explicit layerwise bandwidth control, 2) by modifying initialization scale it makes the spectral supports of layers more or less disjoint. The benefit of doing this is that it enables coupling with standard multigrid / frequency marching approaches to solve inverse problems. Thanks to the introduced modifications the low-frequency content of the function implemented by the network need not change when fitting higher frequencies. The authors show how this leads to high-quality reconstructions of molecules in synthetic CryoEM experiments.

**Questions:**

- What happens with your Cryo-EM reconstructions if you initialize orientations randomly? How important is the current coarse initialization?
- How exactly do you choose the hyperparameters $\lambda$ and epochs / scale count? You state that for PDB4ake you find a better reconstruction if you dedicate more epochs to the first and second scales. Do you do this by comparing the reconstruction to the ground truth or simply eyeing the reconstruction?
- In Cryo-EM, how do other implicit representations (non-marching) perform?

**Limitations:**

The discussion of limitations is adequate.

**Strengths And Weaknesses:**

The problem addressed by the paper is very clear, the paper is well written, the numerical experiments seem well executed and the results are compelling.

The interventions in the architecture and training are simple. They yield solid (but not staggering) improvements over earlier work. Cryo-EM is an important application but the setting treated here is somewhat simplified (no shifts, no heterogeneity, only moderate noise SNR = 0.1, and a coarse initial orientation estimate). It would be interesting to see how the proposed architecture fares on other problems. problems.

As far as I can tell the paper is technically correct—there are no formal results, the architecture description is clear and simple, and the successful experiments strongly corroborate the correctness of the derivations.

In terms of the narrative, I have the impression that achieving perfect scale orthogonality is a bit overstated. It can be certainly play a role in inverse problems like Cryo-EM which are linear (or conditionally linear when given the particle orientations) and where the frequencies don’t mix (by the Fourier slice theorem). (I doubt that it need be ideal.) It may however be a liability or at least not as useful in problems with strong scale coupling.

---

> ### Author Response · Authors · 2022-08-02
> **Response to Reviewer bz3B**
>
> Thank you for the thoughtful review and great questions. Below, we address the main questions and concerns raised in the review:
>
> **“...but the setting treated here is somewhat simplified”**
> - **“no shifts,”**  We omitted particle translation, instead assuming centered particles, for computational expedience, as it speeds up pose estimation.  That said, the maximum likelihood approach we use for pose estimation can be extended to handle in-plane translation straightforwardly.  As such, our conclusions about the residual MFN do not depend on this, but we can also include in-plane translation upon request.
> - **“no heterogeneity,”** The experiments in the paper focus on the homogeneous case for computational expedience. Extensions to the heterogeneous case are typically solved using Expectation-Maximization, for which the homogeneous case is, effectively, done in the M-step. Beyond the simpler workflow, there is nothing in the use of residual MFNs for the homogeneous case that should not also work in the heterogeneous case.  Indeed, it is worth noting that we use a real-space MFN representation instead of the conventional Fourier domain representation in cryo-EM specifically to enable continuous heterogeneity in a manner like 3DFlex [1], although this is out of scope in this paper.
> - **“only moderate noise SNR=0.1,”** The particle images shown in Fig. 4c have noise levels consistent with typical cryo-EM experimental data. The reason the SNR value appears higher than one often sees in experimental settings, is that our images are tightly cropped around the particles (again for computational expediency).  It is well known in the cryo-EM community that SNR depends on the size of the images relative to the size of the particles, since signal power in particle images depends on the spatial support of the particle while the power of the noise increases with the image size.  See Sec. 5.4.2 in [3] for a detailed discussion and visualization of the relation of SNR and image size.
> - **“coarse initial orientation estimate…”** In our experiments, we coarsely initialize poses to have a geodesic distance from ground truth which is uniformly distributed between 45 and 90 degrees (section 6.1). We experimented with a random initialization scheme but found that the optimization is more susceptible to local minima in this case. However, since existing methods can reliably provide initial coarse pose estimates [4-6] we felt it was appropriate to adopt such a coarse initialization scheme for our method.
>
>   Still, we did conduct an experiment in which poses were initialized such that their distance from ground-truth varies uniformly from 0 to 180 degrees. We observed that after allocating more epochs to the first scale and second scales (i.e., 30 epochs each), our method converges to a structure that is on par with the results in the main paper. Specifically, we obtain final resolutions of 2.91A and 2.66A for PDB1ol5 and PDB4ake, respectively.
>
> **“How exactly do you choose the hyperparameters $\lambda$ and epochs / scale count?”**
> - **hyperparameters $\lambda$.** As mentioned in the paper, the lambda hyperparameters control the growth and disjointness of frequency supports. Our first intuition was to avoid settings that create gaps in coverage of the Fourier domain, ie, when $\lambda_2 > 2 + \lambda_1$. Beyond this constraint, we investigated a range of hyper-parameter settings in experiments. In all cases, the network fits the signal well; but we found that for $\lambda_2 = 2$, $\lambda_1 = 0.3$, the model yields a lower test loss (based on held out data for image fitting experiments).
>
>   For cryoEM experiments (3D regression) we found the same hyperparameters perform well, based on L2 distance from the ground truth 3D density. We thus used the same hyperparameters for cryo-EM experiments with 2D images. The discussion of this was abbreviated in the main paper due to space limitations, but we will add more detail and visualization about hyperparameter setting in the supplemental.
> - **Epochs / scale count.** For image fitting experiments, we used 4 scales for fair comparison with BACON. To set the number of epochs for training, we used the behavior of the loss on several held out images. The first two scales fit well with a relatively small number of epochs (500), compared to 8000 epochs for the finest scale. We believe the number of epochs per scale depends on the size and complexity of the target signal; ie, smaller images require fewer epochs at each scale for convergence.
>
>    For cryoEM experiments, the number of epochs for each scale is chosen based on the convergence of the re-rendering loss. For PDB4ake, we found that more iterations are required to obtain accurate pose estimates – We speculate that this is due to more symmetries in the structure, which makes pose optimization more challenging.

---

> > ### Author Response · Authors · 2022-08-02
> > **Response to Reviewer bz3B (Cont.)**
> >
> > **“achieving perfect scale orthogonality is a bit overstated, … at least not as useful in problems with strong scale coupling.”**
> > With residual MFNs one can ensure that the frequencies at one level are disjoint from other levels (illustrated in Fig. 2). Nevertheless, we found empirically that having a small overlap in frequency coverage yields better results. For domains with strong frequency coupling across a wide range of scales, we agree that coarse-to-fine optimization schemes may be suboptimal.
> >
> > **“Other implicit representations (non-marching)”**
> > To the best of our knowledge, previous methods for ab-initio reconstruction with implicit representations [7, 8, 9] all use frequency marching, either explicitly or implicitly. They either employ a hierarchical search scheme, which iteratively refines the poses [8], or the explicitly band-limit the signal during pose inference [7, 9]. Indeed frequency marching is used in all widely used cryo-EM reconstruction algorithms because, without it, local minimas are endemic and highly problematic. We also note that all these previous methods [7,8,9] use frequency domain coordinate networks, while we represent structure in real-space and apply frequency marching for free by being able to directly control the band-limit of the signal representation. As an ablation, we also conducted an experiment where we supervised the network only at the full-scale reconstruction. The right panel of Fig 6 shows poses trapped in local minima, with no improvement in the reconstructed structure for multiple iterations.
> >
> > **References.** \
> > [1] A. Punjani, D.J. Fleet, 3D flexible refinement: structure and motion of flexible proteins from Cryo-EM, BioRxiv. (2021). \
> > [2] S.H. Scheres, Processing of structurally heterogeneous cryo-EM data in RELION, Methods in Enzymology. 579 (2016) 125–157. \
> > [3] R.M. Glaeser, E. Nogales, W. Chiu, Single-particle Cryo-EM of biological macromolecules, IOP Publishing, 2021. \
> > [4] A. Punjani, J.L. Rubinstein, D.J. Fleet, M.A. Brubaker, cryoSPARC: algorithms for rapid unsupervised cryo-EM structure determination, Nature Methods. 14 (2017) 290–296. \
> > [5] J. Banjac, L. Donati, M. Defferrard, Learning to recover orientations from projections in single-particle cryo-EM, ArXiv Preprint ArXiv:2104.06237. (2021). \
> > [6] R. Lian, B. Huang, L. Wang, Q. Liu, Y. Lin, H. Ling, End-to-end orientation estimation from 2D cryo-EM images, Acta Crystallographica Section D: Structural Biology. 78 (2022). \
> > [7] A. Levy, F. Poitevin, J. Martel, Y. Nashed, A. Peck, N. Miolane, D. Ratner, M. Dunne, G. Wetzstein, Cryoai: Amortized inference of poses for ab initio reconstruction of 3d molecular volumes from real cryo-em images, ArXiv Preprint ArXiv:2203.08138. (2022). \
> > [8] E.D. Zhong, T. Bepler, B. Berger, J.H. Davis, CryoDRGN: reconstruction of heterogeneous cryo-EM structures using neural networks, Nature Methods. 18 (2021) 176–185. \
> > [9] E.D. Zhong, A. Lerer, J.H. Davis, B. Berger, CryoDRGN2: Ab initio neural reconstruction of 3D protein structures from real cryo-EM images, in: Proceedings of the IEEE/CVF International Conference on Computer Vision, 2021: pp. 4066–4075.

---

### Meta-Review · Area_Chair_DtoX · 2022-09-02

**Recommendation:** Accept
**Confidence:** Certain

**Metareview:**

The paper studies Multiplicative Filter Networks, which are coordinate neural networks in which each layer applies a multiplicative (Hadamard product) filter and a sinusoidal nonlinearity. The paper shows how introducing residual connections and initializing appropriately can lead to networks where the frequency content of the image separates over layers. This leads to a learned version of classical “coarse-to-fine” reconstruction methods, which the paper terms Residual Multiplicative Filter Networks. The paper illustrates its proposals with experiments on image approximation and on cryo-EM reconstruction.

Reviewers found that the paper presents a simple idea, which can be easily adopted whenever a coarse-to-fine reconstruction is desired, and as such is likely to see followup work. The main questions concerned the necessity of a coarse-to-fine approach in applications where one ultimately seeks a reconstruction at just a single scale, and the cryo-EM experiments, which show good performance compared to a baseline, when the coordinate network model is integrated into a larger system. Overall, the reviewers found that paper presents a natural modification to MFNs which improves both their interpretability and applicability in inverse problems in imaging.


**Award:**

No

---

### Decision · Program_Chairs · 2022-09-14

Accept